# Global Tonga tsunami explained by a fast-moving atmospheric source

R. Omira[1,2 ✉], R. S. Ramalho[3,4,5], J. Kim[1], P. J. González[6,7], U. Kadri[8], J. M. Miranda[1,2], F. Carrilho[1] & M. A. Baptista[2,9]

Volcanoes can produce tsunamis by means of earthquakes, caldera and flank collapses, pyroclastic flows or underwater explosions[1–4]. These mechanisms rarely displace enough water to trigger transoceanic tsunamis. Violent volcanic explosions, however, can cause global tsunamis[1,5] by triggering acoustic-gravity waves[6–8] that excite the atmosphere–ocean interface. The colossal eruption of the Hunga Tonga–Hunga Ha'apai volcano and ensuing tsunami is the first global volcano-triggered tsunami recorded by modern, worldwide dense instrumentation, thus providing a unique opportunity to investigate the role of air–water-coupling processes in tsunami generation and propagation. Here we use sea-level, atmospheric and satellite data from across the globe, along with numerical and analytical models, to demonstrate that this tsunami was driven by a constantly moving source in which the acoustic-gravity waves radiating from the eruption excite the ocean and transfer energy into it by means of resonance. A direct correlation between the tsunami and the acoustic-gravity waves' arrival times confirms that these phenomena are closely linked. Our models also show that the unusually fast travel times and long duration of the tsunami, as well as its global reach, are consistent with an air–water-coupled source. This coupling mechanism has clear hazard implications, as it leads to higher waves along land masses that rise abruptly from long stretches of deep ocean waters.

Volcanic activity has long been recognized as a source of tsunamis, with volcanic earthquakes, gravitational and caldera collapses, pyroclastic flows, underwater explosions and volcanic blasts constituting the main triggers of volcanic tsunami waves[3,4]. Typically, most of these mechanisms do not displace enough water to result in long and far-reaching tsunamis. Consequently, they trigger localized point-sourced tsunamis that dissipate energy very efficiently with distance. However, the generation of interoceanic volcanic tsunamis, although rare, is possible through air–water coupling[9], as illustrated by the approximately CE200 Taupō (New Zealand)[8], the 1883 Krakatau (Indonesia)[1,5,10–12] and the 1956 Bezymianny (Kamchatka)[13] eruptions and associated tsunamis.

Violent explosive eruptions may trigger atmospheric perturbations by a sudden ejection, at supersonic speeds, of lava and gases into the air[14]. If the explosive pressure is sufficiently large, the thrusting of volcanic products into the atmosphere can produce acoustic-gravity waves[15]. Acoustic-gravity waves are low-frequency compression-type sound waves propagating under gravity at speeds close to those of the sound of the medium (for example, about 340 m s$^{-1}$ in air and 1,500 m s$^{-1}$ in water)[16]. As such, they can travel substantial distances before dissipating[14]. Acoustic-gravity waves propagate into different media, becoming hydroacoustic (in water), Scholte (in seabed) or Rayleigh–Lamb (in elastic layers)[17,18]. Critically, acoustic-gravity waves propagating over the

ocean can force the sea surface to generate tsunami-like waves, known as volcano-meteorological tsunamis[4,7]. By their nature, they can also transfer energy into the water by means of resonance[19], particularly when propagating over sufficiently long stretches of deep ocean[20]. This process provides a unique air–water-coupling mechanism that is, in principle, capable of causing exceptionally fast-travelling and far-reaching volcanic tsunamis.

Although air–water coupling has been recognized as a viable mechanism for the generation of tsunamis, all known cases in which eruption-triggered atmospheric oscillations led to transoceanic or global tsunamis predate modern, worldwide dense instrumental networks. Consequently, notable gaps remain in our knowledge of the air–water-coupling processes, the potential magnification of tsunamis through resonance with the atmospheric waves and the true reach of such coupled waves away from their source. Consequently, the full hazard extent of atmospheric-driven tsunamis triggered by volcanic activity is still unknown and, therefore, rarely considered in the design of tsunami early-warning systems.

The colossal explosion of the Hunga Tonga–Hunga Ha'apai volcano in the South Pacific was a source of both noticeable atmospheric waves[21] and an exceptionally fast-travelling global tsunami with minimal dissipation in the far field. We explain this tsunami by a moving atmospheric source

[1]Instituto Português do Mar e da Atmosfera (IPMA), Lisbon, Portugal. [2]Instituto Dom Luiz (IDL), Faculdade de Ciências, Universidade de Lisboa, Lisbon, Portugal. [3]School of Earth and Environmental Sciences, Cardiff University, Cardiff, UK. [4]Lamont-Doherty Earth Observatory, Columbia University, Palisades, NY, USA. [5]Instituto Dom Luiz (IDL) e Departamento de Geologia, Faculdade de Ciências, Universidade de Lisboa, Lisbon, Portugal. [6]Volcanology Research Group, Department of Life and Earth Sciences, Instituto de Productos Naturales y Agrobiología, Consejo Superior de Investigaciones Científicas (IPNA-CSIC), La Laguna, Canary Islands, Spain. [7]COMET, Department of Earth, Ocean and Ecological Sciences, University of Liverpool, Liverpool, UK. [8]School of Mathematics, Cardiff University, Cardiff, UK. [9]Instituto Superior de Engenharia de Lisboa (ISEL), Instituto Politécnico de Lisboa (IPL), Lisbon, Portugal. ✉e-mail: rachid.omira@ipma.pt

mechanism that forces the ocean surface and accompanies both the formation and the propagation of tsunami waves. Crucially, the profusion of atmospheric and sea-level readings that recorded this event worldwide and the availability of geostationary satellite observations of the propagating acoustic-gravity waves provide a unique opportunity to unravel the most enigmatic aspects of air–water coupling and tsunami generation through ocean forcing and resonance. The Tonga tsunami hence represents the first opportunity to investigate the physical mechanism of formation of global tsunamis by acoustic-gravity waves, allowing us to move beyond a 'proof of principle' into the development of useful forecasting models.

## Tonga eruption and acoustic-gravity waves

The 15 January 2022 volcanic explosion at the Hunga Tonga–Hunga Ha'apai volcano in the Kermadec-Tonga intraoceanic volcanic arc[22] is one of the largest in the last 30 years, and possibly even represents a new class of eruptive style, for which no recent precedent is known[23]. Hunga Tonga–Hunga Ha'apai is a very recent volcanic cone that was formed in 2014–2015 as the result of an eruption that connected the older Hunga Tonga and Hunga Ha'apai islands, the only subaerial parts of the larger and active Hunga submarine volcano[24,25]. The latest eruptive phase started in mid-December 2021, when the volcano once again awakened to produce vigorous shallow-water explosive activity. This eruptive style alternated with periods of relative calm for most of the first 12 days of January 2022, during which few explosions were recorded. Then, on 13 and 14 January, shallow-water explosions resumed, disrupting the existing cone[22]. These explosions, however, would be dwarfed by a colossal but very-short-lived explosion (or series of explosions)[26,27] on 15 January, starting a few minutes before 5:10 p.m. local time (4:10 a.m. UTC; see Fig. 1). The explosion was reportedly heard as far away as Alaska and produced one of the tallest eruptive columns of the satellite age, at 35–54 km high, resulting in an umbrella cloud more than 650 km in diameter at its maximum extent[27]. One of the most striking features of this explosion was, however, the unusual pattern of concentrically propagating atmospheric acoustic-gravity waves it created (Fig. 1). These waves propagated from the ocean surface to the ionosphere and then travelled radially outwards across the world several times[21,27]. Reports quickly emerged that the Tonga archipelago was hit by tsunami waves with runup heights up to 15 m and, later, other locations as far as Japan and Chile with wave amplitudes of around 1.0 and 1.5 m, respectively, as corroborated by tide-gauge data from across the Pacific.

## Exceptional tsunami

The tsunami triggered by the Hunga Tonga–Hunga Ha'apai eruption was exceptional, as it was recorded at a global scale, exhibiting higher propagation speeds, unexpected wave heights in the far field and an unprecedented duration. Hence we focus our analysis on the intriguing far-field and global characteristics of this event. The analysis of a wealth of sea-level data comprising a total of 277 records from 230 tide gauges and 47 DART buoys (Extended Data Table 1 and Supplementary Information Table 1) confirms the global reach of the tsunami waves, which propagated at a faster speed than well-known earthquake-triggered and point-sourced tsunamis. Effectively, the globally recorded tsunami arrived much earlier than what would be expected from a standard point-sourced tsunami located at Hunga Tonga–Hunga Ha'apai volcano (Fig. 2a and Extended Data Fig. 1). A comparison of travel times between a hypothetical point-sourced tsunami and the observed tsunami (Fig. 2a and Extended Data Fig. 1) shows that the latter travelled 1.5 to 2.5 times faster, crossing the Pacific, the Atlantic and the Indian oceans in less than 20 h. This difference in the propagation speed is mainly noticeable in the far field (Fig. 2a and Extended Data Fig. 1). For example, the tsunami reached the coasts of Japan and Chile in less than 7 and 10 h, respectively, far exceeding the expected travel time from a point-sourced tsunami (that is, 10.5 to 12.5 h for Japan and 12 to 17 h for

Chile) (Fig. 2a). In the Atlantic and Indian oceans, the tsunami was even faster, such as arriving at the Caribbean in 10 to 11 h, whereas a point-sourced tsunami would take more than 26 h to travel from the source area into the Caribbean, around the South American continent (Fig. 2a). The same applies to the eastern coasts of the Atlantic and the Mediterranean Sea, where ocean disturbances were observed after 16.5 h in Portugal and 17.5 h in Italy (Fig. 2a). Here the travel time from a point source predicts arrivals after 27 and 32 h, respectively. Our analysis suggests that the Tonga tsunami propagated at a speed of about 1,000 km h$^{-1}$, showing no marked decrease in speed from deep to shallow waters, as expected for earthquake-triggered and point-sourced tsunami waves given that their phase speed ($c$) is dependent on the water depth ($h$) over which they propagate (that is, $c = \sqrt{gh}$, in which $g$ is the gravity acceleration).

Sea-level data also reveal unexpectedly large wave amplitudes at distant coastal areas (Fig. 2b) and exceptional tsunami duration (Fig. 2c and Extended Data Fig. 3). Unlike the recent earthquake-triggered devastating tsunamis (for example, the 26 December 2004 tsunami in the Indian Ocean[28] and the 11 March 2011 tsunami in the Pacific Ocean[29]), the tsunami that followed the Hunga Tonga–Hunga Ha'apai explosion shows a longer duration of sea-surface disturbances (more than 1.5 days at most locations of the Pacific and Atlantic oceans, see Fig. 2c and Extended Data Fig. 3) and a minimal dissipation of wave amplitudes across the oceans (Fig. 2b). A direct comparison of sea-surface records at different oceans (Fig. 2b) supports the negligible change in tsunami wave amplitude with increasing distance from the volcano. At a distance of around 67 km, the tide gauge of Tonga Island recorded a maximum wave amplitude of 1.14 m (Fig. 2b and Supplementary Information Table 1), which, however, does not reflect the full near-field tsunami impact as high runup heights (around 15 m) were also observed in Tongatapu Island, Tonga archipelago, suggesting the contribution from a local tsunami source (probably of volcanic or gravitational origin). Notably, at the far field, comparable or even higher values were observed within a distance of about 10,000 km, at sea-level stations in Arica (1.18 m) and Coquimbo (1.43 m) in Chile (Fig. 2b). Moreover, a point-sourced tsunami model (Extended Data Fig. 2) suggests that these distant coasts should not have been reached by any tsunami disturbances. Furthermore, the observed tsunami did not experience a noticeable amplitude dissipation during its interoceanic propagation, as the maximum wave amplitudes of about 50 cm recorded in Hawai'i are comparable with those observed in the Caribbean and the eastern Atlantic coast (Portugal) (Fig. 2b). A point-sourced tsunami also fails to propagate across the oceans, as the numerical simulation shows no waves reaching the Indian or Atlantic oceans (Extended Data Fig. 2).

## Tsunami of atmospheric origin

The global reach of the tsunami, its high propagation speed, its long duration and its minimal wave dissipation in the far field call for a moving source-generation mechanism that accompanies the tsunami propagation and continuously pumps energy into it, rather than a localized (point) source. A comparison of the atmospheric and oceanic data shows a direct correlation between the first passage of the air-pressure disturbance and the onset of the tsunami in many distant locations around the globe (Fig. 2c). This suggests that the observed far-field tsunami waves—or at least the first recorded waves—are the direct response of the ocean surface to the air-pressure disturbance forcing. Moreover, the latter clearly coincided with the first passage of the acoustic-gravity wave that resulted from the volcanic explosion of 15 January 2022 (Fig. 2d), supporting the atmospheric (acoustic-gravity wave) origin of the Tonga tsunami. The barometric pressure records also allow the identification of up to two more main air-pressure disturbances (Fig. 2c, Extended Data Fig. 4 and Supplementary Information Video 1) that we interpret as the second and third passages of the acoustic-gravity wave after propagating across the globe (Fig. 2c). This

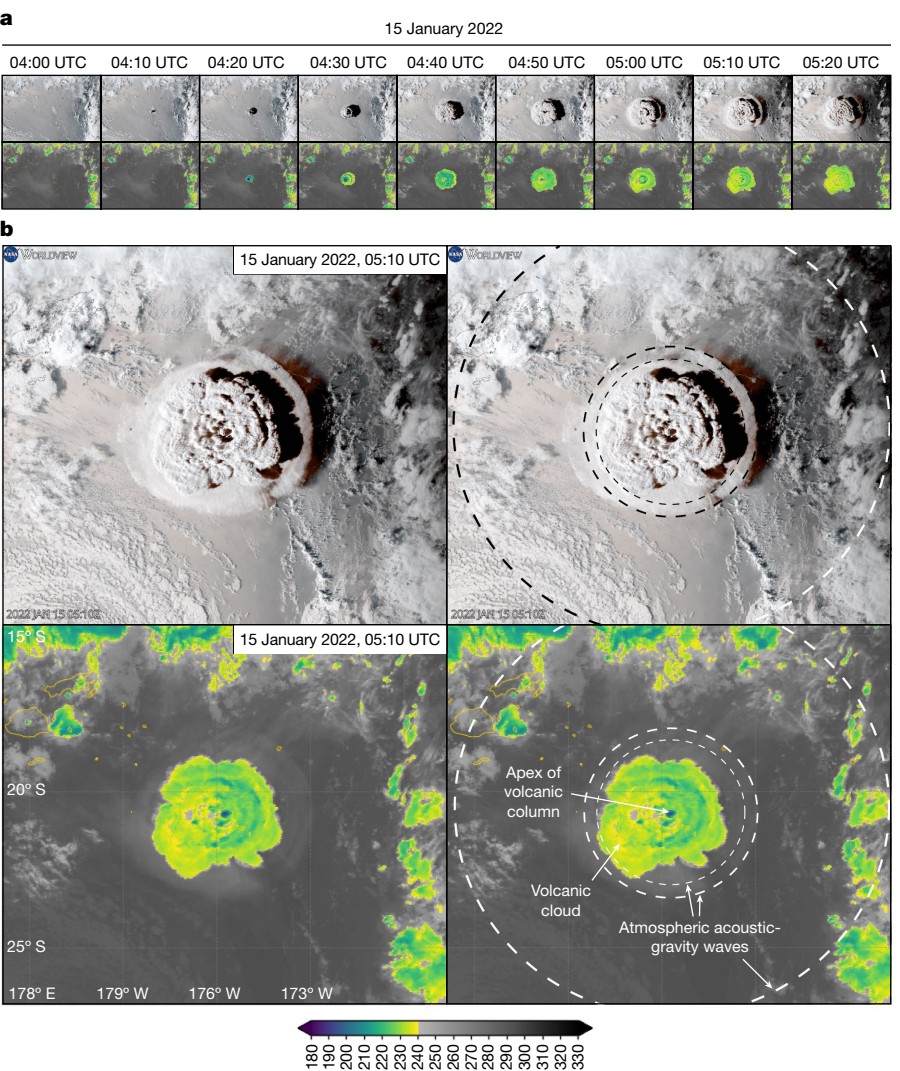

**Fig. 1 | Satellite imagery of volcanic source. a**, Optical (top) and infrared channel 13 (bottom) from the GOES-17 satellite imagery showing the onset and expansion of the Hunga Tonga–Hunga Ha'apai volcanic explosion and cloud, from 04:00 to 05:20 UTC. Optical imagery from NOAA/NESDIS/STAR visualized by NASA Worldview. **b**, Detail at 5:10 UTC, with clean and interpreted optical (top pair) and clean and interpreted infrared channel 13 (bottom pair) imagery. Note the visible (although faded) effects of the rapidly expanding acoustic-gravity waves triggered by the colossal explosion of the volcano.

explains the unusually long duration of the tsunami observed in most locations, as each arrival of the air-pressure disturbance leads to the reinitiation of the ocean disturbance.

The interpretation that the globally observed tsunami was of atmospheric origin and driven by acoustic-gravity waves is quantitatively supported by a tsunami numerical model forced by a moving air-pressure disturbance. Critically, this model—which is based on a finite volume method solver of the non-linear shallow-water equations equipped with air-pressure forcing terms (Methods and Extended Data Fig. 5)—is able to reproduce the tsunami's fast propagation (Fig. 3a), long duration (Fig. 3b and Supplementary Information Video 2) and global reach (Fig. 3c and Supplementary Information Video 2). The model also closely reproduces the arrival time observations (Fig. 3a and Extended Data Fig. 1), in stark contrast to the point-sourced tsunami propagation, which fails to explain the early arrival of the first tsunami waves (Extended Data Fig. 1). Moreover, it provides a valid mechanism to explain the interoceanic wave propagation, showing that waves were rapidly generated across the oceans, giving the impression that these waves 'jump' from one ocean to another (for example, from the Pacific to the Atlantic oceans across Central America) with a minimal loss of wave height (Fig. 3b and Supplementary Information Video 2). A simulation covering 36 h of temporal propagation (Supplementary Information Video 2) shows that at each passage of the air-pressure disturbance—in the direction away from or back to the source—the ocean surface is further excited and new waves are generated (Fig. 3b), explaining the long duration of the observed tsunami. The model also explains how the atmospheric-driven tsunami amplifies through a resonance mechanism, leading to localized sizeable waves in areas adjacent to the deep-water oceanic trenches (Fig. 3c). It closely reproduces, at many open-ocean locations, both the observed maximum wave amplitudes (Fig. 3d) and the recorded time series (Extended Data Fig. 6), but with certain discrepancies (Methods). Ultimately, the simulated global distribution of the tsunami maximum wave amplitudes shows minimal interoceanic wave dissipation, as is particularly noticeable in the Atlantic Ocean (Fig. 3d).

## Tsunami mechanism

On the basis of these observations, analyses and simulations, we propose a model for the generation and propagation of the Hunga

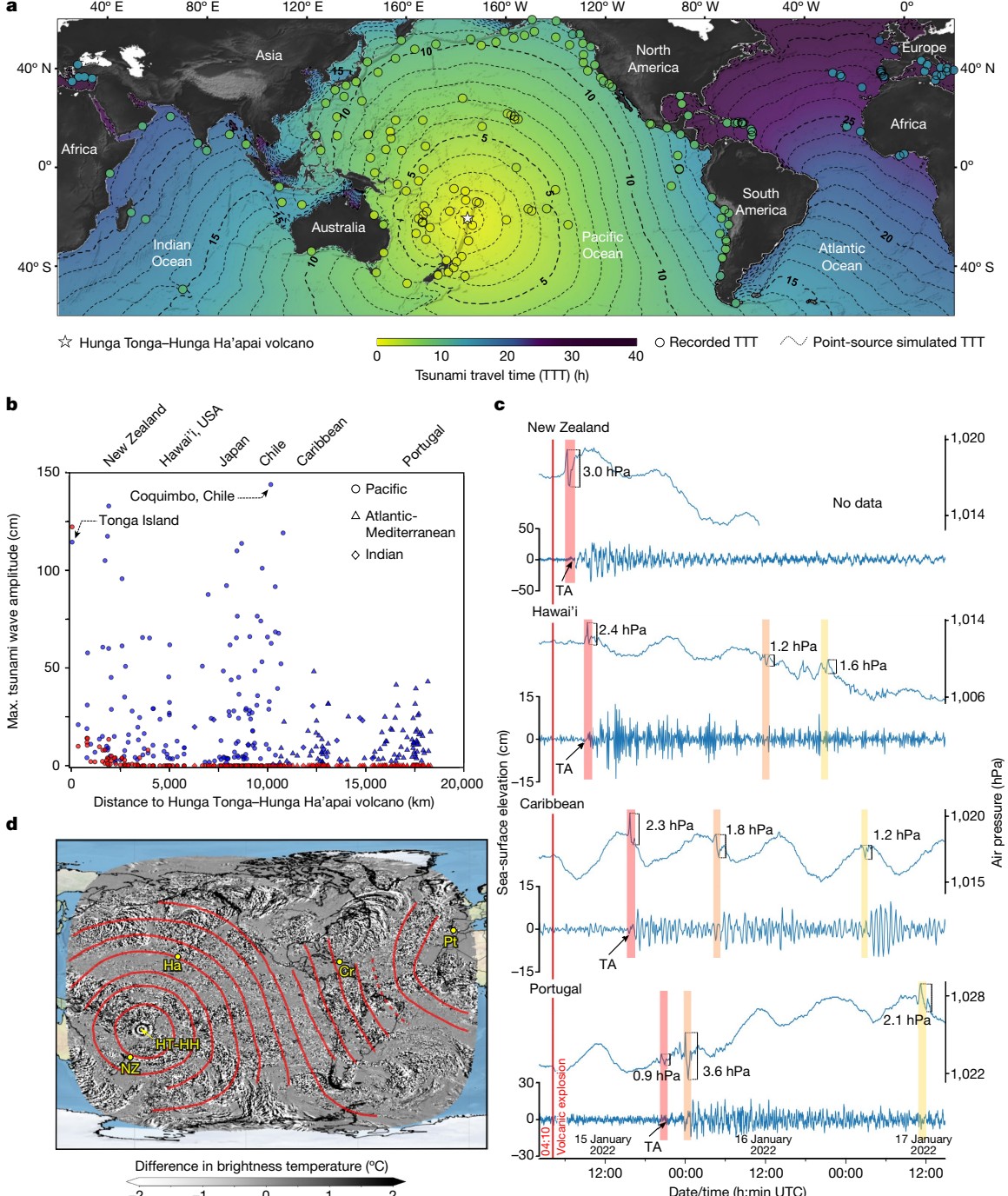

**Fig. 2 | Characteristics of the Hunga Tonga–Hunga Ha'apai tsunami as derived from sea-level, atmospheric and satellite data. a**, Observed tsunami travel times from sea-level records (coloured dots) compared with those from a standard point source located at the Hunga Tonga–Hunga Ha'apai (HT-HH) volcano (background colour). **b**, Variation of the maximum recorded tsunami wave amplitudes (blue symbols) as a function of the horizontal distance from the source and comparison with maximum wave amplitudes predicted by a point-sourced tsunami model (red symbols) (Extended Data Fig. 2). **c**, Correlation of the tsunami arrival times (TA) with the passage of the air-pressure disturbances (red bars) recorded at weather stations in New

Zealand (NZ), Hawai'i (Ha), the Caribbean (Puerto Rico, Cr) and Portugal (Pt); the red bar highlights the passage of the first air-pressure disturbance, whilst the orange and yellow bars are, respectively, interpreted as the second and third passages. **d**, Hourly travel times (red contours starting at 6:00 UTC) of acoustic-gravity waves identified from satellite images showing the difference in top-of-atmosphere brightness temperature in °C (Supplementary Information Video 1). Note that the passage of the acoustic-gravity wave over NZ, Ha and Cr (red contours in **d**) correlates well with the time in which the air-pressure disturbances are recorded at those stations (red bars in **c**).

Tonga–Hunga Ha'apai eruption-triggered tsunami, and more generally for tsunamis generated by acoustic-gravity waves resulting from violent volcanic explosions (Fig. 4). In this model, both the generation and propagation aspects of the observed tsunami are set by the physical characteristics of the acoustic-gravity wave resulting from

the sudden mass injection and vertical compression of the air column above the volcanic explosion. At the air–water interface, the explosion can be pictured as a flow through a circular opening or a distribution of sources over the volcanic column cross section. Thus, owing to wave diffraction (Huygens–Fresnel principle) and density

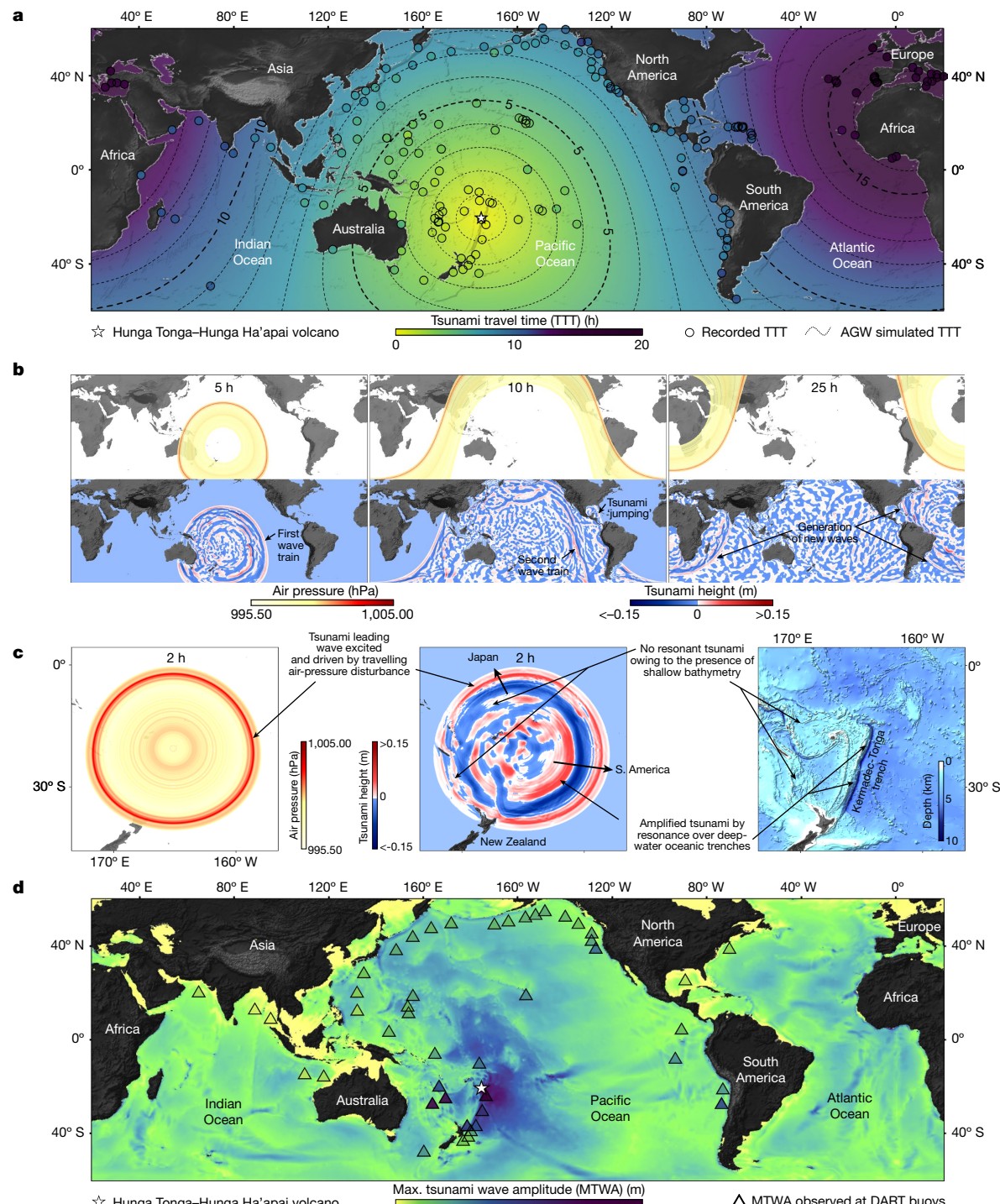

**Fig. 3 | Numerical model for the Hunga Tonga–Hunga Ha'apai global tsunami driven by a moving air wave. a**, Observed tsunami travel times from sea-level records compared with those from a tsunami driven by a moving acoustic-gravity wave (AGW). **b**, Temporal propagation of the AGW (top panels) and the evolution of the subsequent tsunami (bottom panels) (snapshots from the 36-h temporal propagation of AGW and tsunami waves, Supplementary Information Video 2). **c**, Pattern resulting from the tsunami amplification process through resonance occurring between the moving air-pressure disturbances and the tsunami waves over the deep water of the Kermadec-Tonga trench. **d**, Global distribution of the simulated maximum tsunami wave amplitudes compared with those observed at open-ocean stations (DART buoys).

stratification of the air and refraction, the generated acoustic-gravity waves propagate mainly tangentially to the ocean surface. We interpret the first arrival of the tsunami as a direct response of the ocean surface to the passage of the air-pressure disturbance, which results in the generation of forced surface water waves under the inverse barometer effect. Accordingly, the first tsunami wave train (Figs. 3b

and 4) explains the earlier arrival of the tsunami across the globe with respect to a point-sourced tsunami (Extended Data Fig. 1), as the acoustic-gravity waves travel much faster than the ocean-gravity waves in intermediate to shallow ocean water depths. The second tsunami wave train to arrive is, by contrast, linked to resonant ocean-gravity waves with acoustic-gravity waves. Such a mechanism occurs when the

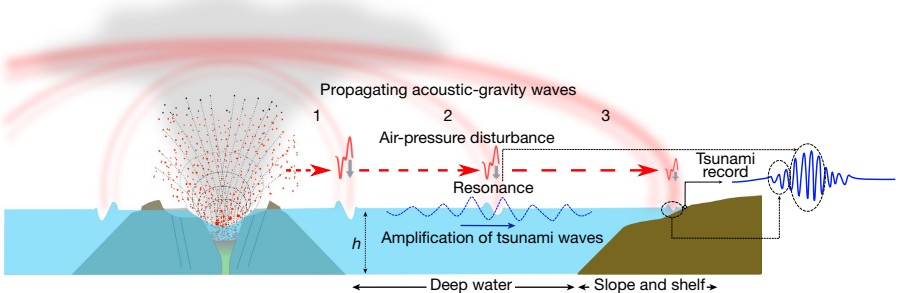

**Fig. 4 | Model for the Hunga Tonga–Hunga Ha'apai tsunami, as driven by the acoustic-gravity waves generated by the volcanic explosion (or series of explosions).** Numbers 1, 2 and 3 correspond to progressive moments in time. In 2, the dashed blue signal corresponds to the resonant tsunami propagating under gravity towards the coast. The tsunami record on arrival to the shoreline will exhibit a first component resulting from the arrival of the air-pressure disturbance, followed by oscillations resulting from the amplification of the tsunami waves by means of resonance with acoustic-gravity waves in the open deep ocean.

subsequent main envelope of the acoustic-gravity wave interacts with the first tsunami wave train (Methods and Extended Data Figs. 7 and 8), resulting in an air–water energy transfer that leads to an increase in tsunami wave amplitudes (Extended Data Fig. 8). For sinusoidal waves of identical wavelength and frequency, almost all of the initial energy can be transferred between the two types of waves, whereas for wave packets, the interaction becomes less efficient, with only up to 40% of the initial energy being transferred[19]. The generated tsunami can also amplify under Proudman resonance[30], when the acoustic-gravity wave speed matches the speed of the ocean long waves in very deep water, leading to higher-amplitude waves (Fig. 3c) that propagate under gravity towards the coast, that is, explaining the second tsunami wave train of later arrival (Fig. 3b), as recorded around the world by around 100 coastal stations, among those here analysed. This resonance mechanism is, however, not expected in shallow waters, given that tsunami waves become slower and do not resonate with acoustic-gravity waves in these areas (Fig. 3c), thus explaining why land masses surrounded by wide and shallow continental shelves did not experience sizeable (or, in some cases, even noticeable) tsunami amplitudes. Crucially, the highest tsunami amplitudes and runup heights are expected along land masses that rise abruptly from the abyssal plains or oceanic trenches, as occurs at oceanic islands and coastlines adjacent to subduction zones. In these settings, the amplification of the resonant tsunami waves with acoustic-gravity waves is maximized and minimally attenuated before impact ashore. This explains why notable tsunami amplitudes were recorded at distant coasts such as Japan (about 1 m) and along the western margin of South America (about 1.43 m in Chile) (Fig. 2b and Supplementary Information Table 1). In ocean basins, comprising both shallow and deep waters such as in the Caribbean and the Mediterranean, tsunami amplification is believed to occur in deep-water areas, as shown by the numerical model (Fig. 3c,d), and then the amplified waves propagate towards the shallow-water coastal areas, reaching around 20–50 cm at some locations (Fig. 2b and Supplementary Information Table 1). Atmospheric-driven tsunamis of volcanic origin are, thus, different from earthquake-triggered and point-sourced tsunamis in terms of generation, propagation and impact, with clear hazard implications: they travel much faster, experience minimal energy dissipation as a function of increasing distance, can reach the size of earthquake-triggered tsunamis as enough energy gets pumped into them and tend to pose a much larger threat to land masses that rise abruptly from the deep ocean, in a clear contrast to latter tsunamis that amplify in shallow-water areas.

In summary, our data analysis (Figs. 1 and 2 and Extended Data Fig. 1), numerical modelling (Fig. 3, Extended Data Fig. 1 and Supplementary Information Video 2) and analytical model (Extended Data Figs. 7 and 8) provide a consistent and quantitative interpretation of the exceptional aspects of the tsunami that followed the Hunga Tonga–Hunga Ha'apai volcano colossal explosion. Our findings demonstrate that acoustic-gravity waves radiating from powerful volcanic explosions may constitute a moving source that transfers energy into the ocean by means of resonance, resulting in fast-travelling, far-reaching and enduring high tsunamis (Fig. 4). This global tsunami triggered by acoustic-gravity waves sourced by a volcanic eruption–the first to be recorded with modern, globally dense instrumentation–also brings to the fore the need to revisit the forecast capabilities of the tsunami early-warning systems in place at present. Our study provides further insights on the source mechanism of previously identified rare global volcanic tsunamis occurrences, such as the one that followed the 1883 colossal eruption of Krakatau, and has implications for the hazard potential of such events.

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

## Methods

### Satellite images processing

Earth observation from satellites captured the build-up and consequences of the Hunga Tonga–Hunga Ha'apai eruption and explosion, as it affected the atmosphere[23], ocean, that is, sea-surface discoloration or thermal anomalies, and the island morphological changes[31,32]. To map the rapid evolution associated with this volcanic explosion, we relied on atmospheric weather satellites that capture the state of the atmosphere at high temporal resolution (every 10 min). Here we collected Advanced Baseline Imager Level 2 Full Disk Cloud and Moisture Imagery (ABI-L2-CMIPF) product from geostationary GOES-16 and GOES-17 satellites, infrared channel, channel 13 = 10.3 μm. The downloaded and analysed data spanned the period from 15 January 2022 to 20 January 2022 through AWS (https://docs.opendata.aws/noaa-goes16/cics-readme.html#about-the-data). The ABI-L2-CMIPF product provides top-of-atmosphere brightness temperature in kelvin (Fig. 1). Pressure-induced changes in temperature are visible as travelling waves in the subsequent imagery from 15 January 2022 04:10 UTC. To highlight the transient signals, we differentiate the ABI-L2-CMIPF products with a 10-min delay and obtain changes in temperature (Supplementary Information Video 1). We observed that transient-temperature difference signals are a good indicator—within the satellite temporal sampling rate of 10 min—of the position of the front of the explosion travelling vertically averaged atmospheric pressure barometric waves (Fig. 2c). We used this approach to track the position of the acoustic-gravity wave at different times (see contour lines in Fig. 2d).

### Oceanic and atmospheric data analysis

We collected both oceanic and atmospheric data to help understand the propagation of the Hunga Tonga–Hunga Ha'apai tsunami and its origin. The oceanic data consist of sea-level records from 709 (out of 824) tidal stations available on the IOC web platform (http://www.ioc-sealevelmonitoring.org), from 20 tide gauges and 37 DART buoys of NOAA (https://www.ndbc.noaa.gov) and from ten DART buoys of NIWA (https://niwa.co.nz/). This dataset was then enriched by local data from the Portuguese, Spanish and French coastal tide gauge networks (total of 52 sea-level readings). The atmospheric data comprise a total of 80 barometric pressure records obtained from different sources and agencies (NOAA in the USA, NIWA in New Zealand, and IPMA in Portugal). Extended Data Table 1 summarizes the specifics of the collected data and their sources.

A careful quality check and processing of the collected sea-level data were undertaken. They include sorting the data by date/time, selecting the relevant time frame, detecting outliers and spikes, and then removing them without affecting the data quality. To harmonize the dataset, sea-level readings were converted to the same measuring unit (millimetres), averaged over a 1-min sampling interval and then linearly interpolated to fill the data gaps.

The quality of the air-pressure data, with sampling intervals between 1 and 10 min, was checked in a similar manner to the sea-level data. Data were cleaned from missing values and when the sampling intervals were not uniform, the interval was determined by using the most common time difference between two samples in a dataset and rounding it to the nearest minute. Air-pressure measurements were then converted to hPa.

Once the quality check of the data was completed, the sea-level time series were detided and filtered, using an order 5 Butterworth high-pass filter, to keep only waves in the frequency range of tsunamis. Stations were then selected on the basis of the location and data quality to capture as much as possible the features of the tsunami event. Sea-level stations with air-pressure data—allowing for solid air–water correlations—were preferred. For these data, the maximum wave height (crest to trough), the maximum wave amplitude and the start time of the event were determined. The start time was attributed by visual inspection of

the data and, to examine the consistency of the models with observation (that is, point-sourced and atmospheric tsunami models), we only relied on arrival times with very low uncertainty in what concerns the tsunami onsets (Figs. 2a and 3a). This selection of stations (total of 277) and the corresponding analysis are depicted in Supplementary Information Table 1. Examples of sea-level and air-pressure data analyses are depicted in Extended Data Figs. 3 and 4.

### Tsunami numerical model

**Reconstruction of air-pressure forcing.** As our work focussed on the far field, we reconstructed the air-pressure disturbance around the Hunga Tonga–Hunga Ha'apai volcano from barometric signals recorded relatively far away from the source, that is, the high-resolution (1-min) pressure data available in New Zealand. To achieve this, we assumed that the air-pressure disturbance propagates radially and we used the observation at Kaitaia, New Zealand as an input (Extended Data Fig. 5a). This choice of referral point allows the estimation of the initial total energy of around $3.4 \times 10^{23}$ ergs, according to the formula in ref. [33] (equation (1)). This energy estimate was then used to simulate the tsunami generation from a point-source model (Extended Data Fig. 2a). Our estimate also indicates that the explosion of the Hunga Tonga–Hunga Ha'apai volcano released less energy (about 2.5 times) than the Krakatau volcano in 1883, which was estimated to be $8.6 \times 10^{23}$ ergs (ref. [33]).

$$E = \frac{2\pi R_0 H \sin(d_0/R_0)}{\rho_0 V} \int P_0^2 \mathrm{d}t, \tag{1}$$

in which $E$ is the energy, $R_0$ is the earth radius, $H$ is the height of the homogeneous atmospheric layer, $\rho_0$ is the air density, $V$ is the sound speed, $P_0$ is the observed air-pressure disturbance at distance $d_0$ away from the volcano and $t$ is the duration of the disturbance.

We assumed that the air-pressure disturbance propagated at a constant speed $v_0$. To estimate $v_0$, we performed numerical simulations with pressure speed between 300 m s$^{-1}$ and 350 m s$^{-1}$ and compared the results with the arrival times of the maximum air-pressure disturbance at six stations (Marshall Islands, Apra, Sand Island, Hilo Bay, Jeju Island and Charlotte in the Caribbean). From these comparisons, we obtained the optimal speed value of 322 m s$^{-1}$, as it shows the best agreement in air-pressure disturbance arrivals between the simulations and observations (error less than 0.042, see Extended Data Fig. 5b).

As an input, $AP_{obs}(t)$ denotes the filtered record of the air-pressure disturbance of Kaitaia at time $t$ (Extended Data Fig. 5a). Let $d_0$ be the distance from the volcano to the observation point and $v_0$ the speed of the air-pressure disturbance (=322 m s$^{-1}$ in this work).

Suppose we want to find the atmospheric pressure $AP(t, d)$, in which $t$ is the elapsed time after the volcano eruption and $d$ is the distance from the volcano. Then the atmospheric pressure is given by Taylor's formula as,

$$AP(t, d) = AP_{obs}(t + (d_0 - d)/v_0) \sqrt{C_1/\sin(d/R_0)} \tag{2}$$

in which $R_0$ is the earth radius and $C_1$ is a constant. The constant $C_1$ is chosen to ensure the match of the air pressure at the referral point (here, Kaitaia, New Zealand, see Extended Data Fig. 5a). However, we observed that $AP(t, d)$ goes to infinity at the volcano (or $d = 0$) and its antipode. For this reason, we used a relaxation method as follows:

$$AP(t, d) = AP_{obs}(t + (d_0 - d)/v_0) \sqrt{C_1/(\epsilon_0 + \sin(d/R_0))} \tag{3}$$

with a small constant $\varepsilon_0$. Last, to account for the energy dissipation—here assumed as exponential—we applied the following equation:

$$\begin{aligned} AP(t, d) &= AP_{obs}(t + (d_0 - d)/v_0) \sqrt{C_1/(\epsilon_0 + \sin(d/R_0))} \\ &\cdot \exp(-\lambda_0 t/2) \end{aligned} \tag{4}$$

in which $(\varepsilon_0, \lambda_0) = (0.005, 1 \times 10^{-5})$ allows the accurate reproduction of most air-pressure records (example of Marshall Islands and Hilo Bay, Hawai'i, Extended Data Fig. 5c).

Equation (4) allows the derivation of the air-pressure time series at each location (each distance from the Hunga Tonga–Hunga Ha'apai volcano) including at the Hunga Tonga–Hunga Ha'apai volcano itself (Extended Data Fig. 5d), which was then used as a forcing condition for ocean disturbances. Examples of air-pressure time series at some selected distances from the Hunga Tonga–Hunga Ha'apai volcano are depicted in Extended Data Fig. 5d.

**Tsunami numerical code.** We used the GeoClaw numerical code[34,35], a finite volume method solver of the non-linear shallow water equations, to simulate the propagation of the tsunami waves following the explosion of the Hunga Tonga–Hunga Ha'apai volcano. To account for the air-pressure wave as a trigger and driver of the ocean waves propagation, the GeoClaw code was equipped with atmospheric pressure forcing terms following the governing equations presented in ref. [36]. The non-linear shallow water governing equations were then solved spatially and temporarily in a system of spherical coordinates, accounting for the effect of the Coriolis force, Proudman resonance[30] and implementing the moving atmospheric forcing following equation (4).

We performed tsunami numerical simulations over a uniform bathymetric grid spacing (1-arc-minute horizontal resolution), extending from 60° S to 60° N and 180° W to 180° E and covering the globe's oceans and seas where the tsunami was observed/recorded. The numerical model was run for a time window of 36 h of propagation to ensure a better representation of both the air-pressure wave travelling twice across the globe and the global reach of the tsunami it caused (Supplementary Information Video 2).

We undertook a rigorous validation of the tsunami model. The lack of high-resolution coastal bathymetry however limited the tsunami validation process using sea-level records at tide gauges. For this reason, we relied on the DART buoys' tsunami records to assess the performance of our numerical solution. Extended Data Figure 6 depicts the comparison between the simulated and the recorded tsunami waveforms at various locations of the Pacific, Atlantic and Indian oceans. Overall, the numerical simulation results fairly reproduce the signals recorded at the open-sea DART stations, particularly in terms of the tsunami arrival time and the first wave (Extended Data Fig. 6). However, they show some discrepancies in terms of the late-arrival-wave amplitudes. We believe that the quality (that is, horizontal resolution) of the bathymetric data used here strongly affects the modelling results, leading to such inconsistency between the simulated and recorded waveforms. Another aspect that is not covered here and might have also influenced the modelling results is the dispersive behaviour of the waves propagating away from the source. Moreover, some sea-level records might also include tsunami waves that are possibly triggered by a secondary point source of volcanic or gravitational origin.

## Analytical model
We considered two layers of ideal compressible homogeneous fluids of thickness $h_j$, in which $j = 1, 2$ denote the top (air) and bottom (water) layers, respectively. Both layers reside in a gravitational field of constant $g$ and are treated as inviscid barotropic fluids each with constant speed of sound $c_j = \sqrt{dp/d\rho_j}$, and the motion is assumed to be irrotational. We defined small non-dimensional parameters $\mu_j = gh_j/c_j^2$, in which $\mu \ll 1$ and the speed of sound in air and water, $c_1 = 340$ m s$^{-1}$ and $c_2 = 1{,}500$ m s$^{-1}$, respectively, both far exceed the maximum phase speed of surface-gravity waves. We considered $h_1/c_1$ as the timescale and $\mu h_1$ as the length scale following ref. [19]. On the basis of irrotationality, the problem is formulated in terms of velocity potential $\phi(r, z, t)$, in which $\mathbf{u} = \nabla\phi$ is the velocity field assuming radial symmetry. The field equation governing $\phi$ in the fluids is obtained by combining the continuity equation with the unsteady Bernoulli equation[37,38],

$$\phi_{tt} - \frac{1}{\mu^2}\left(\phi_{rr} + \frac{1}{r}\phi_r + \phi_{zz}\right) + \phi_z + |\nabla\phi|_t^2 + \frac{1}{2}\mathbf{u}\cdot\nabla(|\nabla\phi|^2) = 0. \quad (5)$$

In our analysis, we introduced two independent models, one for the generation and the other for the propagation of acoustic-gravity waves. All indices will be omitted unless necessary.

**Generation of acoustic-gravity waves.** For the generation of acoustic-gravity waves, we considered the air layer as the sole layer, with the interface assumed to be rigid. We take a cylindrical coordinate system with the $z$-axis vertically upwards and the origin is at the interface, at a midpoint of the sudden vertical motion of an air column of cylindrical geometry following ref. [39],

$$\phi_z = wH(R^2 - r^2)H[t(2\tau - t)], (z = 0), \quad (6)$$

in which $H$ is the Heaviside step function, $R$ is the radius of the disturbance (explosion) and $w$ and $\tau$ are its vertical velocity and half the duration, respectively. At the free surface ($z = 1/\mu$), the combined free-surface condition can be obtained after expanding the kinematic and dynamic conditions, which is given by:

$$\phi_{tt} + \phi_z = 0, (z = \frac{1}{\mu}), \quad (7)$$

Attention is focused on the linear aspects of the problem in the far field only; therefore, the far-field wave equation (5) reduces to the form,

$$\mu^2\phi_{tt} - \left(\phi_{rr} + \frac{1}{r}\phi_r + \phi_{zz}\right) = 0, \left(0 < z < \frac{1}{\mu}\right). \quad (8)$$

Applying separation of variables on equations (6), (7) and (8), Fourier transform of the velocity potential, then inverse Fourier transform, a solution for the non-dimensional pressure at the interface induced by the generated acoustic-gravity wave modes is obtained (for the detailed derivation, see ref. [39]):

$$P(r, t) = 4wR \sum_{n=1}^{\infty} \int_{\omega_{sn}}^{\infty} \frac{\lambda_n \sin(\omega\tau)\sqrt{2/\pi r k_n}}{k_n[2\lambda_n + \sin(2\lambda_n)]}J_1(k_nR)\cos$$
$$\left[k_nr - \omega(t-\tau) - \frac{\pi}{2}\right]d\omega, \quad (9)$$

in which $n$ is the acoustic-gravity wave mode number, $k_n$ is the wavenumber and $\lambda_n = \sqrt{k^2 - \omega^2/\mu^2}$ is the eigenvalue of the dispersion relation, $\omega_{sn} = \pi(n - 1/2)$ is the cut-off frequency and $J_1$ is the Bessel function of the first kind. Note that dimensionally $P = -\rho\phi_t$, in which $\rho$ is the air density, and, thus, the non-dimensional pressure in equation (9) was normalized by $\rho gh$.

An example for the generation of the first acoustic-gravity wave mode ($n = 1$) is given in Extended Data Fig. 7. More specifically, Extended Data Fig. 7 presents calculations of equation (9) for the pressure induced at the air–water interface at Kaitaia, New Zealand. The model successfully predicts the order of magnitude (and, more qualitatively, the shape) of the pressure amplitude arriving during the first hour, although it over-predicts it at a later time frame. This discrepancy is due to the rigidity assumption and the absence of a dissipation mechanism in the model.

**Resonance interaction.** In this part of the analysis, we considered the propagation of acoustic-gravity waves (generated by the linear model above) and their non-linear interaction with the water surface. Note that we treated both layers separately and only allowed coupling through the interface, which is considered here as the free surface for both layers. For simplicity, we neglected stratification of air and assumed

that acoustic-gravity waves reflect completely after a given effective height (at which $\phi_z = 0$). The bottom of the water layer is assumed to be rigid and, thus, $\phi_z = 0$ there as well. At the free surface (interface), expanding equation (7) gives,

$$\phi_{tt} + \phi_z + |\nabla\phi|_t^2 - [\phi_t(\phi_{tt} + \phi_z)]_z + \frac{1}{2}\nabla\phi \cdot \nabla(|\nabla\phi|^2)$$
$$-\frac{1}{2}(\phi_{tt} + \phi_z)(|\nabla\phi|^2 - \phi_t^2)_z - (\phi_t\,|\nabla\phi|_t^2)_z = 0. \tag{10}$$

Resonance is possible between a triad comprising two surface-gravity waves of complex amplitudes $S_\pm$, wavenumbers $k_\pm$ and frequencies $\omega_\pm$ travelling in opposite directions and interacting with an acoustic-gravity mode of amplitude $A$, wavenumber $\mu\lambda$ and frequency $\omega$ (ref. [19]). The triad resonance requires that the dispersion relations are satisfied, along with the conditions:

$$\omega_+ + \omega_- = \omega + \mu\beta,\ k_+ + k_- = \mu\lambda, \tag{11}$$

in which $\beta$ is a tuning parameter. Following ref. [38], the total potential takes the form:

$$\phi = \tilde{\alpha}[A(T)\cos\omega_n(Z+1)e^{i(M\mu\lambda r - \omega t)} + \mathrm{c.c.}]$$
$$+ \epsilon[S_+(T)e^{|k_+|z}e^{i(Mk_+ r - \omega_+ t)} + \mathrm{c.c.}]$$
$$+ \epsilon[S_-(T)e^{|k_-|z}e^{i(Mk_- r - \omega_- t)} + \mathrm{c.c.}] \tag{12}$$

in which $T = \mu t$ is the slow interaction timescale, $\epsilon \ll 1$ and $\tilde{\alpha} = O(1)$ are the steepnesses of the disturbances and c.c. is the complex conjugate. Following similar steps as in ref. [38], although with modified $\tilde{\alpha} \approx \sqrt{\mu_2/\mu_1}$ to account for the different parameters $\mu_j$ involving the two layers instead of a single layer as originally derived, the amplitude evolution equations have the same form:

$$\frac{\partial A}{\partial T} = -\frac{i}{2\omega}\left(\frac{\partial^2 A}{\partial R^2} + \frac{1}{R}\frac{\partial A}{\partial R} + A\right) + \frac{(-1)^n}{8}\tilde{\alpha}\omega^3 S_+ S_-(1 + M^2) \tag{13}$$

$$\frac{\partial S_\pm}{\partial T} = -\frac{(-1)^n}{4}\omega_n\omega^2\tilde{\alpha}AS_\mp^* - \frac{i}{256}\omega^7\tilde{\alpha}^2[(-3 + 8M^2 - M^4)$$
$$|S_\pm^2|S_\pm^*$$
$$-(2 + 4M^2 + 2M^4)|S_\mp|^2 S_\pm] \tag{14}$$

Again, the solution is identical to equations (4.9) and (4.16) of ref. [38] but with the emphasis that the steepness parameters are modified by proper factors. Note that when $M = 1$, the 2D solution of ref. [19] is, in principle, retrieved. Even though the propagation is radial relative to the source, the non-linear interaction is analysed in the far field, at which we set as the origin ($X = 0$). Thus, both positive and negative values of $X$ are allowed, with $+X$ and $-X$ being radially outwards and inwards, respectively. To gain qualitative understanding of the resonant generation of the surface-gravity waves by an acoustic-gravity mode, we consider as an initial condition a Gaussian wave packet with amplitude $A(0) = A_0\exp(-X^2)$ and some small perturbations at the surface, say Gaussian as well. We only focus on the fundamental acoustic-gravity wave mode ($n = 0$), with $\lambda_0 = 1$, $\omega_0 = \pi/2$, $\omega^2 = \pi/4$, $\beta = 0$ and $\tilde{\alpha} = 1$. The non-linear interaction results in the generation of a surface-gravity wave along with the acoustic-gravity wave (Extended Data Fig. 8). In addition, the main envelope of the acoustic-gravity wave interacts with the generated surface-gravity wave that transforms further energy, resulting in amplifying the surface-gravity wave amplitude by about 50%. Note that more than 40% of the energy can be transferred by means of this mechanism[19] for finely tuned conditions. However, because the resonant interaction modulates the waves, energy transfer becomes less efficient as time passes. To further assess the resonance mechanism, the order of magnitude of the peak acoustic-gravity wave pressure can

be obtained from ref. [12]. After restoring dimensions $p = 2\tilde{\alpha}\mu_1^c\omega\,|A|\,\rho_1 c_1^c$ and recalling that $\tilde{\alpha} = \sqrt{\mu_2/\mu_1}$, the peak pressure can be calculated. For example, considering the same parameters as in Extended Data Fig. 8 and in addition substituting $A$ to be on the order of 10 mm, say, 50 mm, and $\omega = 20$ rad s$^{-1}$, both as calculated from the linear model with the parameters for the Kaitaia location, the pressure is found to be on the order of a few hPa, confirming both the linear model and observations.

## Data availability

GOES-16 and GOES-17 satellite imagery data are available at https://docs.opendata.aws/noaa-goes16/cics-readme.html#about-the-data. Optical satellite imagery was collected by the NASA Applied Sciences Program and are publicly available at https://earthdata.nasa.gov/worldview/worldview-image-archive/explosive-eruption-of-hunga-tonga-hunga-ha-apai-volcano. Sea-level datasets are available at the Intergovernmental Oceanographic Commission (IOC) of UNESCO in its Sea Level Station Monitoring Facility (http://www.ioc-sealevelmonitoring.org). DART buoys and US coastal tide-gauge records are available at the Center for Operational Oceanographic Products and Services of the National Oceanic and Atmospheric Administration (NOAA) (https://www.ndbc.noaa.gov). Air-pressure data are available at the Center for Operational Oceanographic Products and Services of the National Oceanic and Atmospheric Administration (NOAA) (https://tidesandcurrents.noaa.gov) and at IPMA (https://www.ipma.pt). Bathymetric/topographic data are available at the General Bathymetric Chart of the Oceans (GEBCO) (https://www.gebco.net/data_and_products/gridded_bathymetry_data/).

## Code availability

The tsunami numerical code GeoClaw is available on the Clawpack website (https://www.clawpack.org/geoclaw.html). The Coupled GeoClaw-atmospheric air-pressure code is available on request from the corresponding author. The satellite-data-processing scripts are available at the following public repository (https://github.com/pablojgonzalez/2022_HungaTongaHungaHaapai). Figure maps were produced with the Generic Mapping Tools (GMT)[40] and the Python libraries cartopy and matplotlib. The codes related to the analytical solutions are available on request from the corresponding author.

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

**Acknowledgements** This work was supported by projects: PTDC/CTA-MET/32004/2017 FAST, funded by FCT—Fundação para a Ciência e Tecnologia, I.P.; PTDC/CTA-GEO/28588/2017 - LISBOA-01-0145-FEDER-028588 UNTIeD, co-funded by the European Regional Development

Fund (ERDF), through Programa Operacional Regional de Lisboa (POR Lisboa 2020), and by FCT; Spain Ministerio de Ciencia e Innovación proyecto PID2019-104571RA-I00 de investigación financiado por MCIN/AEI/10.13039/501100011033 and a 2020 Leonardo Grant for Researchers and Cultural Creators, BBVA Foundation to P.J.G. We would like to acknowledge Instituto Hidrográfico (IH, Portugal) and National Institute of Water and Atmospheric Research (NIWA, New Zealand) for providing sea-level and atmospheric data. We also thank colleagues from IPMA, R. Deus and C. Dutsch (funded by FAST project) for their support in collecting and processing atmospheric and sea-level data, and D. Gamboa for proofreading. Finally, we thank the three reviewers Matthew Alford, Raphaël Paris and Emily Lane for their encouraging comments that helped improve the paper.

**Author contributions** R.O. and R.S.R. initiated, designed and coordinated the project, and wrote the manuscript with key contributions from P.J.G., U.K., J.K. and J.M.M. J.K. and R.O. developed and performed the numerical simulations. R.O., J.K., M.A.B., F.C. and J.M.M. interpreted tsunami simulations. R.O., J.K. and F.C. compiled and analysed sea-level and atmospheric data. R.S.R. and P.J.G. analysed and described the volcanic source. P.J.G. compiled, processed and analysed satellite data. U.K. designed and carried out the analytical model. All authors discussed the data interpretation and commented on the manuscript.

**Competing interests** The authors declare no competing interests.

**Additional information**
**Correspondence and requests for materials** should be addressed to R. Omira.

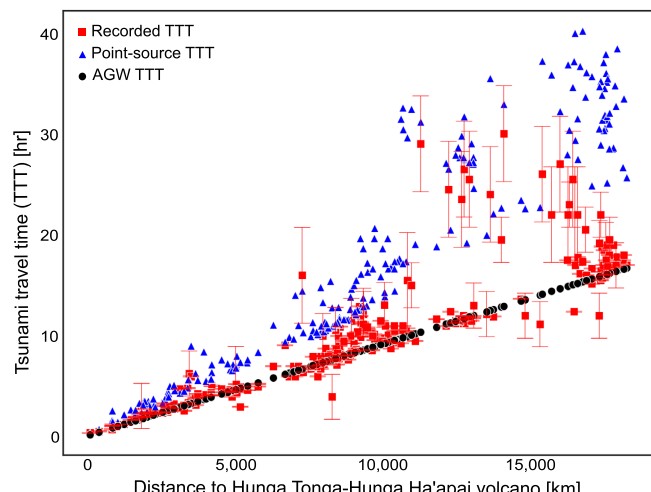

**Extended Data Fig. 1 | Recorded tsunami travel time (TTT) (red squares) compared with TTT from both a point-sourced tsunami (blue triangles) and an acoustic-gravity wave (AGW)-driven tsunami (black circles).** The red bars present the errors in the estimation of the tsunami arrival time from the sea-level records, which are dependent on confidence in determining the tsunami onset through visual inspection (Methods). Here we attributed three error levels: 0.1 h for high confidence, 5 h for low confidence and 10 h for completely uncertain tsunami arrival.

**a**

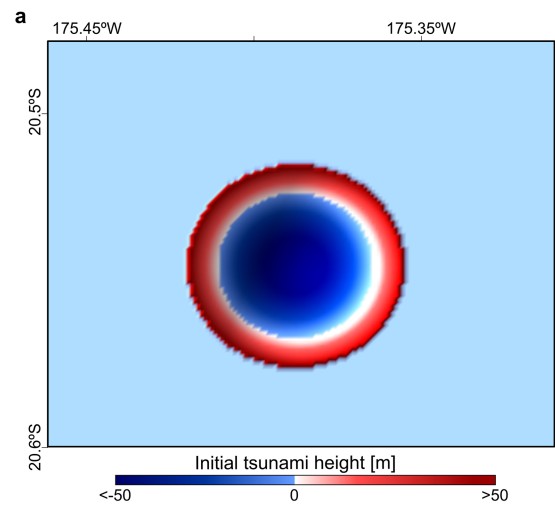

**b**

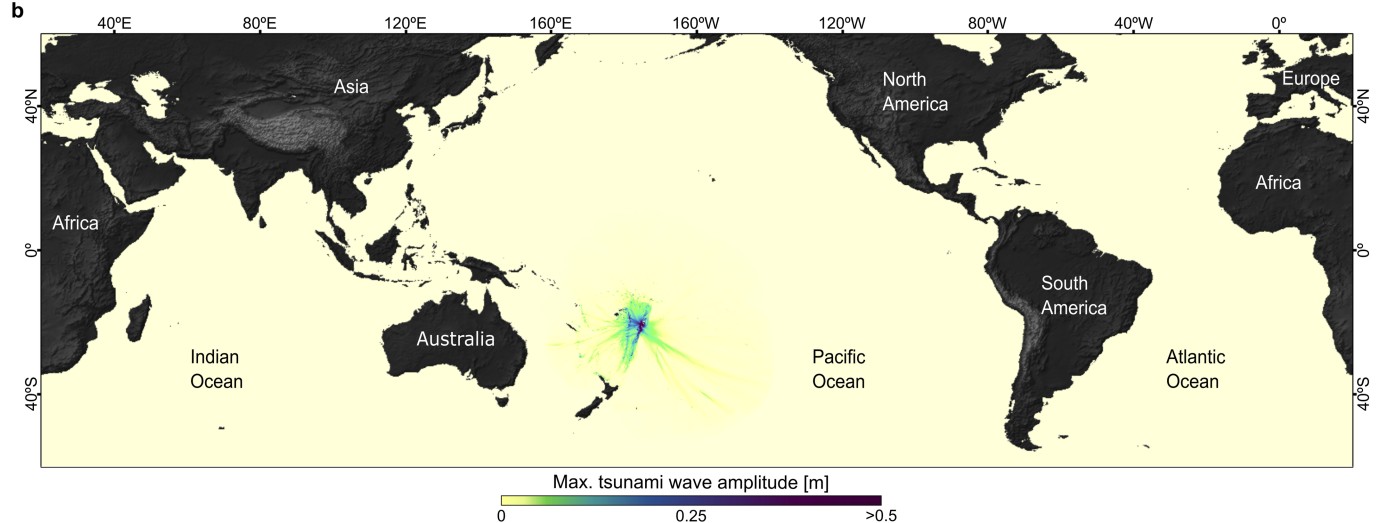

**Extended Data Fig. 2 | Numerical model for a hypothetical point-sourced tsunami triggered by the Hunga Tonga-Hunga Ha'apai underwater volcano explosion. a**, Initial tsunami elevation simulated following the underwater explosion model of ref. [41] assuming a shallow explosion (100–150 m depth) of an estimated energy of about $3.4 \times 10^{23}$ ergs (see Methods). **b**, Numerical simulation of maximum wave amplitudes distribution from a point-sourced tsunami (**a**) using a non-linear shallow water code (refs. [34,35]). The simulation results show marked wave dissipation at distant and transoceanic regions, as no tsunami waves reach the Pacific distant coasts (that is, Japan and South America) or the Atlantic and Indian oceans.

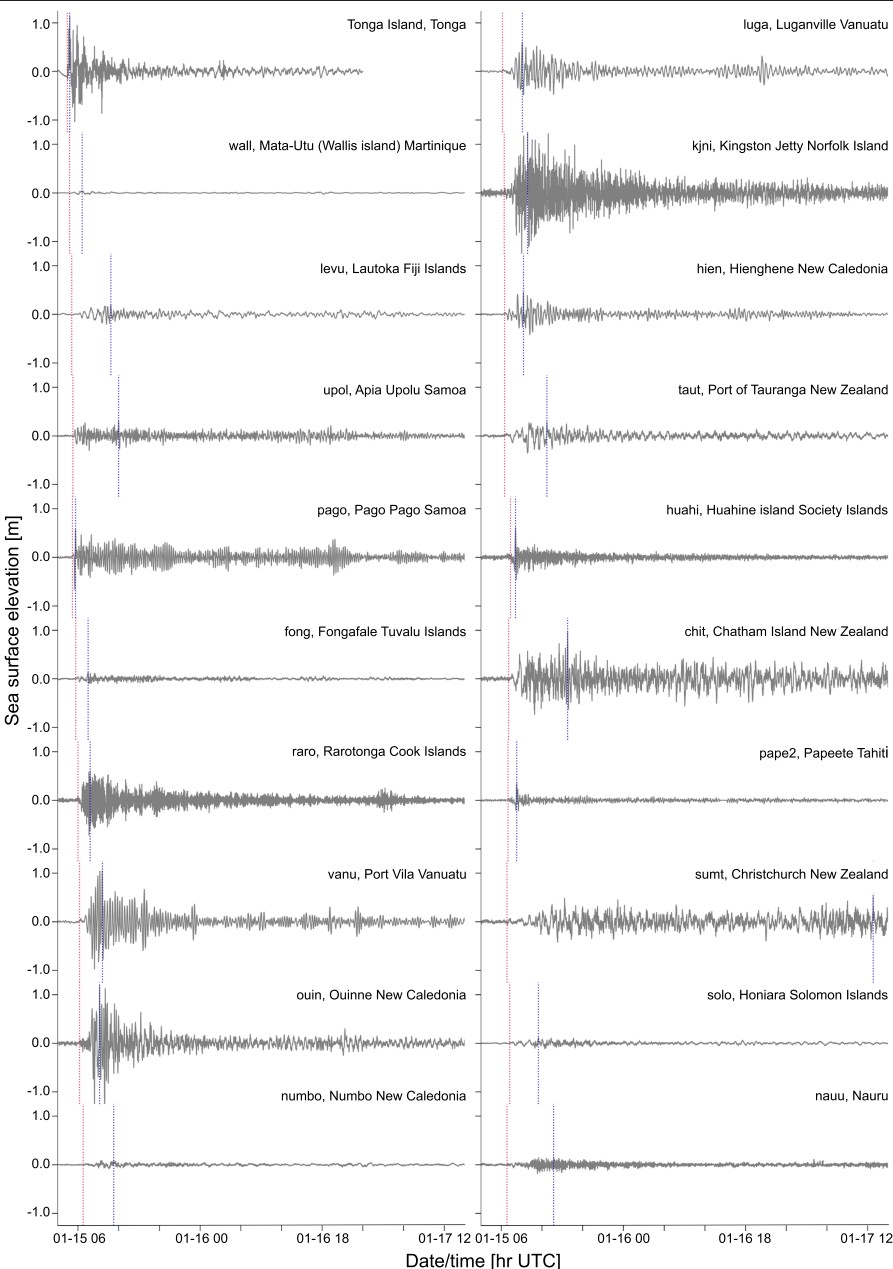

**Extended Data Fig. 3 | Sea-level data analysis.** Detiding, filtering and tsunami arrival time (dashed red line) and maximum wave height (dashed blue line) determination. Examples from the 20 nearest tide-gauge records (see Supplementary Information Table 1).

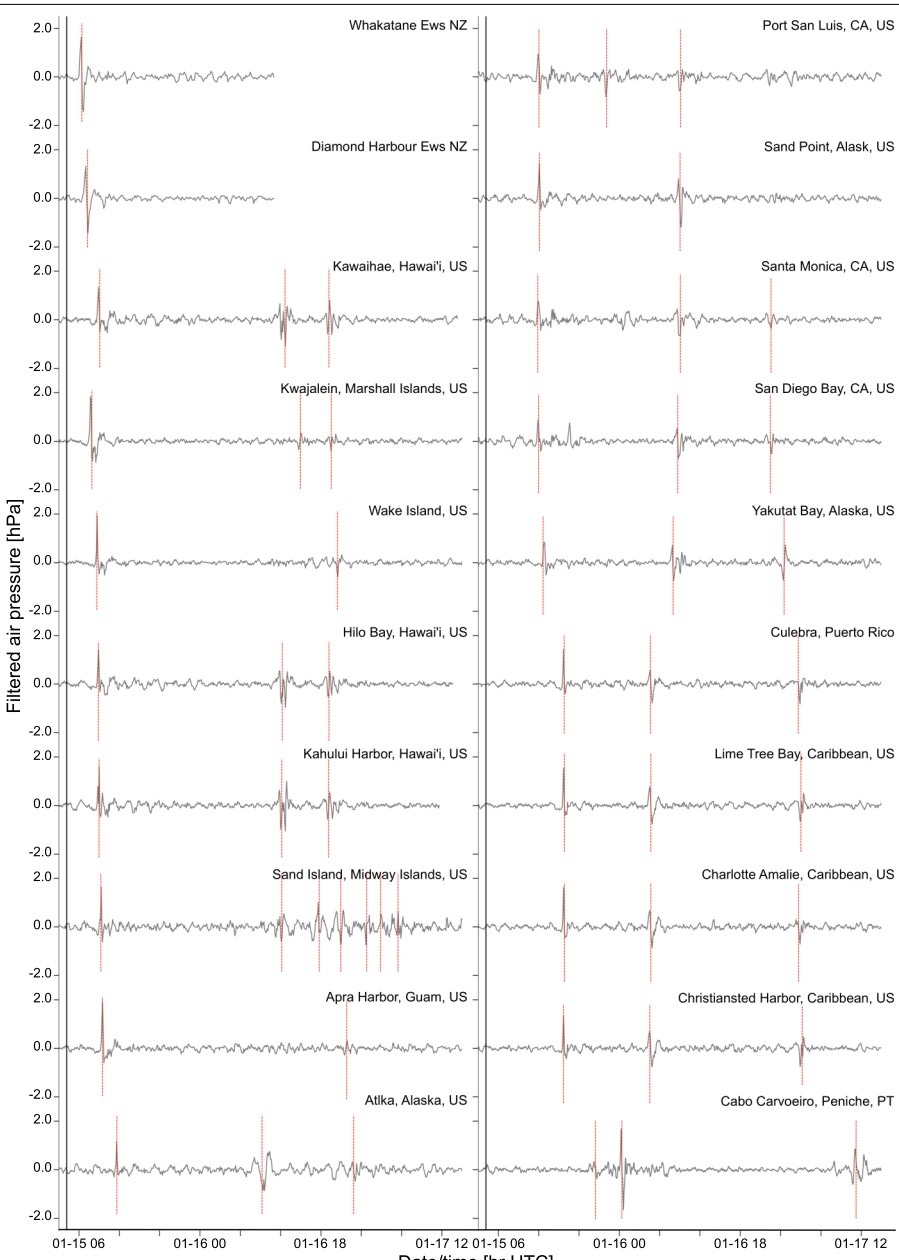

**Extended Data Fig. 4 | Air-pressure data analysis.** Records filtering and determination of main air-pressure disturbances (dashed red lines). Examples from different barometric stations in the Pacific (USA and New Zealand (NZ)) and Atlantic (Portugal (PT)) oceans.

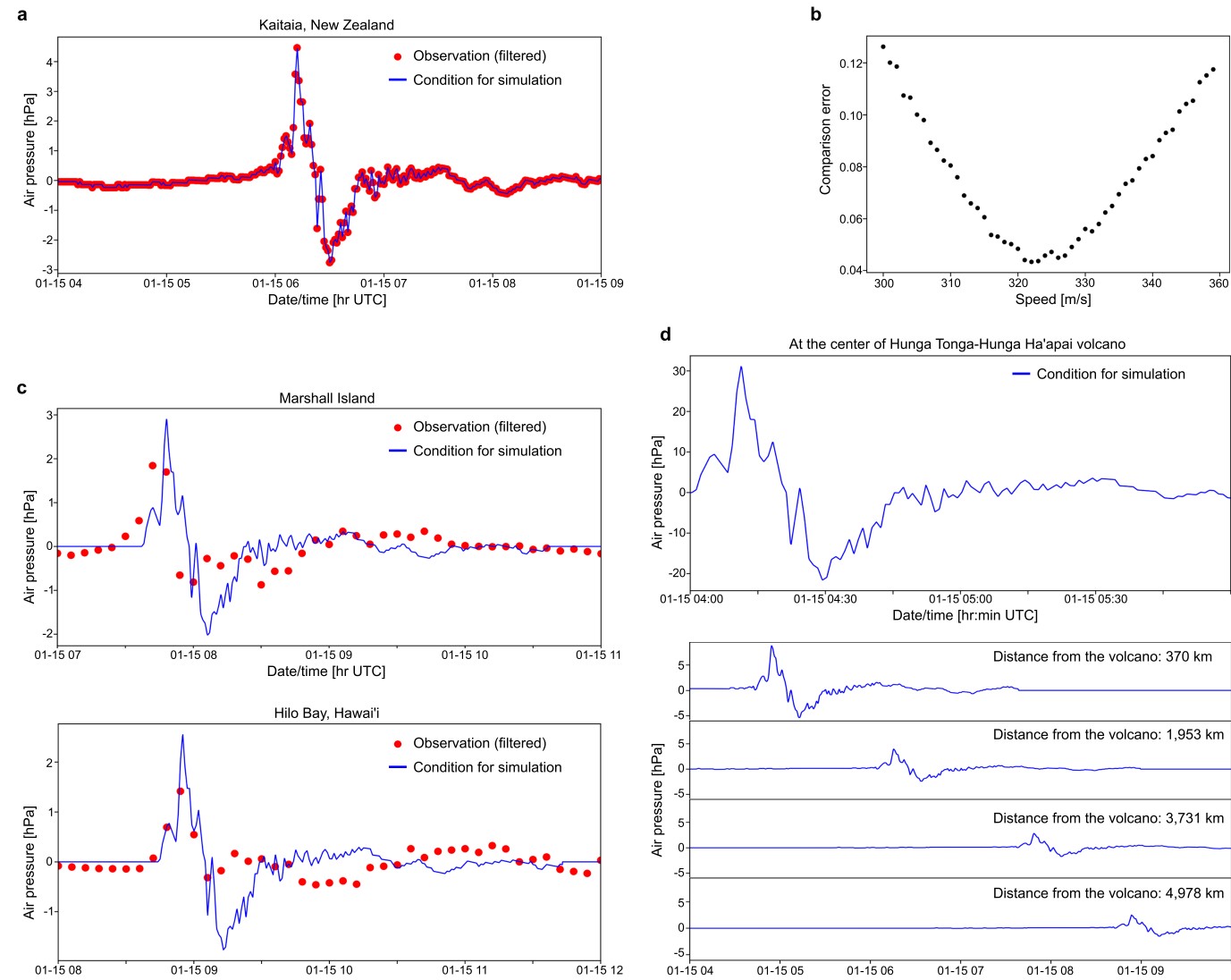

**Extended Data Fig. 5 | Reconstruction of the air-pressure disturbance from high-resolution observations. a**, Comparison between the filtered air-pressure record (red dots) and the model signal (blue line) at the New Zealand observatory. **b**, Estimate of the optimal acoustic-gravity wave speed through the numerical derivation of the comparison errors between the recorded and synthetic arrivals of the air-pressure disturbance for a speed range of 300–350 m s⁻¹. **c**, Reproduction of air-pressure disturbances at Marshall Islands and Hilo Bay, Hawai'i. **d**, Reconstructed air-pressure disturbance at the Hunga Tonga–Hunga Ha'apai volcano (top panel) and at different distances from it (bottom panel).

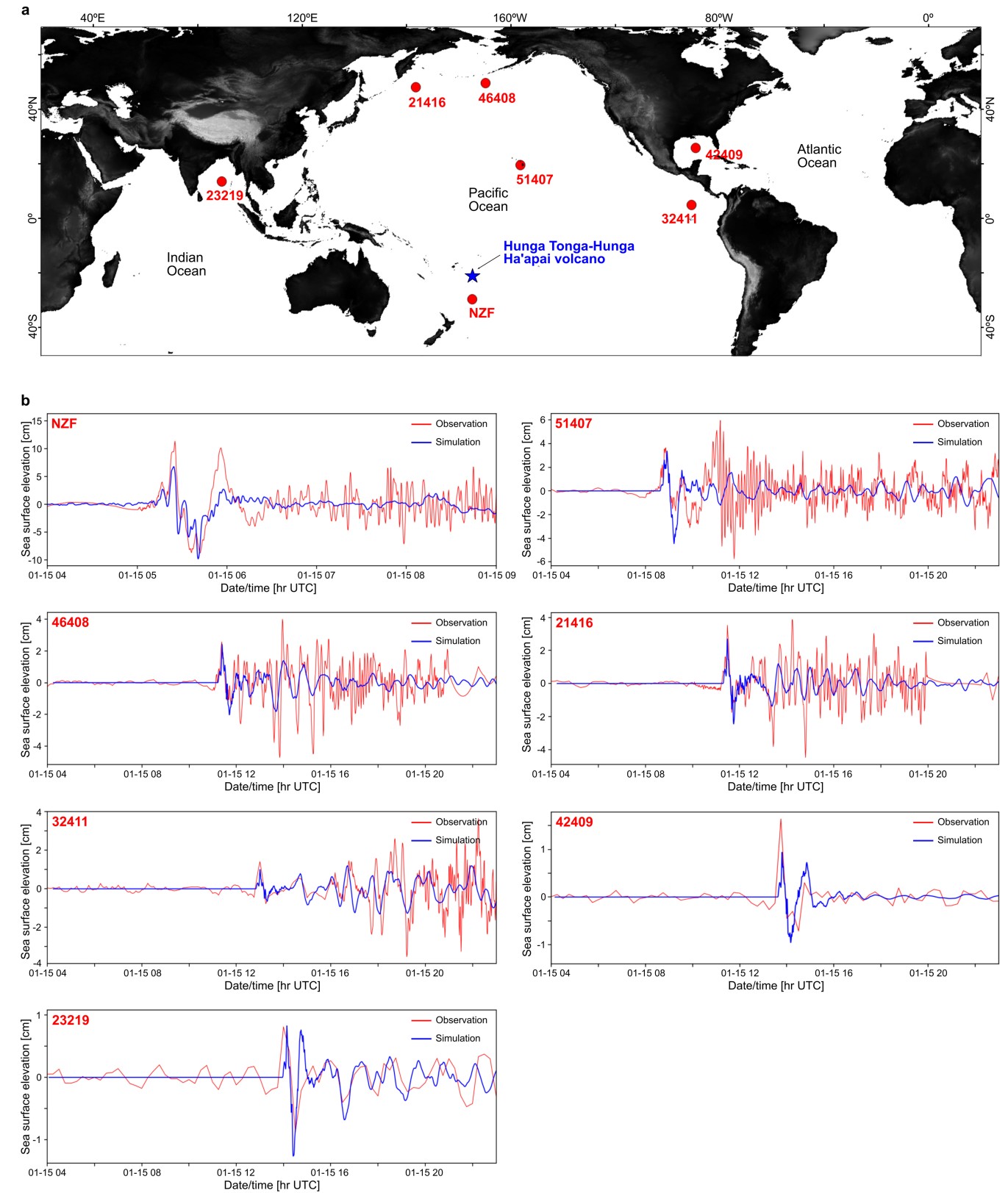

**Extended Data Fig. 6 | Assessment of the performance of the tsunami numerical model forced by a moving air-pressure disturbance. a**, Map of the locations of the open-ocean DART buoys (red circles) where the simulated and recorded time series are compared; numbers and letters in red indicate the station codes for the sea-level buoys (Supplementary Information Table 1). **b**, Comparison of recorded (red signal) and simulated (blue signal) time series at DART buoys located in the Pacific (NZF, 32411, 51407, 46408 and 21416 stations), Indian (station 23219) and Atlantic (Caribbean, station 42409) oceans. The comparison shows a good agreement in the tsunami arrival times as well as in the reproduction of the first wave but with certain discrepancies in the maximum wave amplitudes.

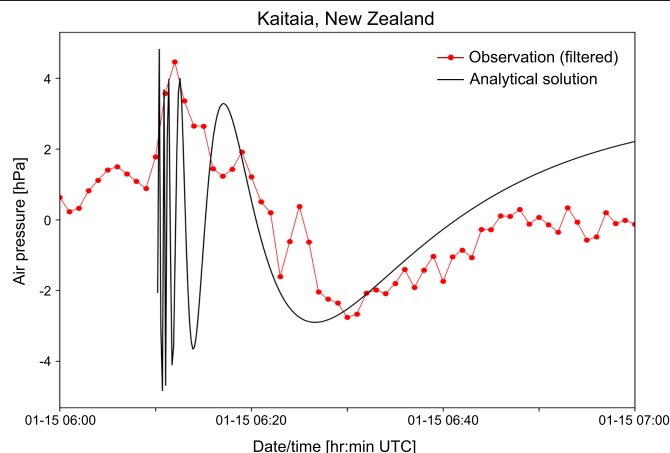

**Extended Data Fig. 7 | Induced air pressure at the interface.** Air-pressure measurements (dotted red line) compared with the analytical solution (black line) of the atmospheric pressure (equation (9)) at the interface induced by the first acoustic-gravity wave mode as it arrives at the New Zealand Kaitaia observatory. With $g = 9.81\ \text{m s}^{-2}$, $c = 343\ \text{m s}^{-1}$, $\rho = 1.2\ \text{kg m}^{-3}$, $n = 1$, $R = 1,100$ m, $h = 500$ m, $r = 1,855$ km, $w = 400\ \text{m s}^{-1}$ and $\tau = 0.2$ s.

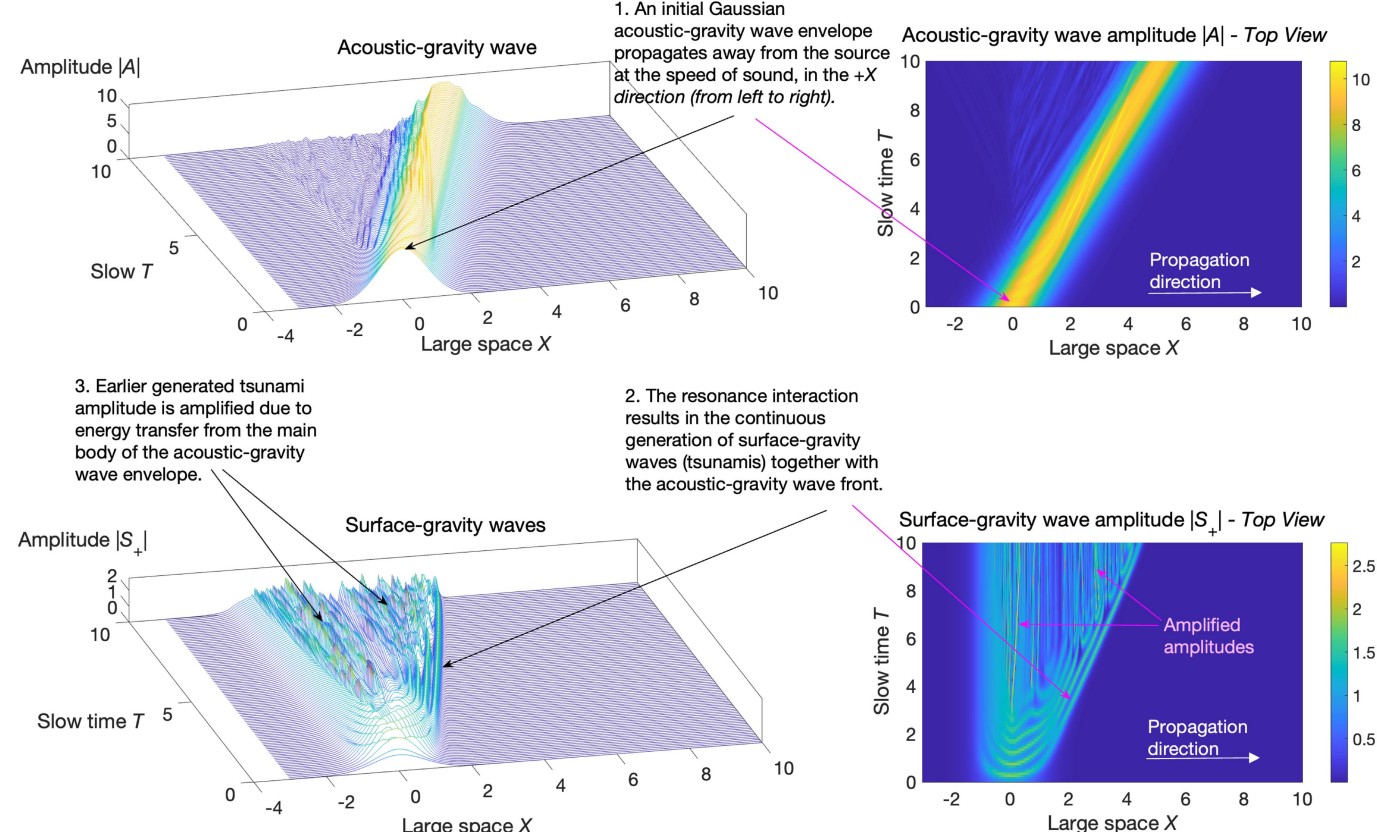

**Extended Data Fig. 8 | Triad resonance non-linear interaction of an acoustic-gravity wave envelope with the water surface.** The envelope of the acoustic-gravity wave is considered Gaussian $A(0) = A_0 \exp(-X^2)$ with a relatively strong initial amplitude of $A_0 = 10$. Here we consider the fundamental mode only with $n = 0$, $\lambda_0 = 1$, $\omega_0 = \pi/2$, $\omega^2 = \pi^2/4$, $\beta = 0$ and $\tilde{\alpha} = 1$. The acoustic-gravity wave generates two surface-gravity waves (that is, tsunami) as it propagates at the speed of sound in air. On the other hand, the main envelope body of the acoustic-gravity wave transfers energy into the previously generated tsunami, which increases its amplitude. Note that the generated surface-gravity waves are similar although not identical, as $S_+$ travels outwards, whereas $S_-$ travels inwards. For clarity purposes, the latter was omitted from the figure.

**Extended Data Table 1 | Collected raw data used in this study**

| Sea-level data | | | | |
|---|---|---|---|---|
| **Origin** | **Area** | **Type** | **Amount** | **Sample rate** |
| IOC - Intergovernmental Oceanographic Commission | Worldwide | Tide-gauge | 709 accurate out of 824 | mostly 1 min |
| NOAA - National Oceanic and Atmospheric Administration | Pacific Ocean and Caribbean | Tide-gauge | 20 | 6 min |
| NOAA | Pacific, Atlantic and Indian Oceans | DART buoys | 37 | 15 min / 1 min |
| NIWA - National Institute of Water and Atmospheric Research | Pacific Ocean | DART buoys | 10 | 15 min / 1 min |
| IH - Instituto Hidrográfico | Portugal | Tide-gauge | 20 | 1 min |
| IPMA - Instituto Português do Mar e da Atmosfera | Spain and France | Tide-gauge | 32 | 1 min |
| **Atmospheric data** | | | | |
| **Origin** | **Area** | **Type** | **Amount** | **Sample rate** |
| NIWA | New Zealand | Air pressure | 2 | 1 min |
| NOAA | Pacific and Caribbean | Air pressure | 20 | 6 min |
| IH and IPMA | Portugal | Air pressure | 20 | 1 min |

Sea-level and atmospheric raw data with indication of their origin, area, amount and sampling resolution.