## [Peer Review File · Nature]

Manuscript Title: Global Tonga tsunami explained by a fast-moving atmospheric source

Reviewer Comments & Author Rebuttals

Reviewer Reports on the Initial Version:

Referees' comments:

Referee #1 (Remarks to the Author):

Please see attached PDF

I summarize what I understand to be the results of this paper here:

-Tsunamis are usually thought to be forced by the direct displacement from land (e.g. new lava, earthquakes and landslides) to ocean. In the recent Tonga quake, we show that this mechanism is inconsistent with the observations (which for the first time are global in scale), in several ways: i) displaced land volume is too small to generate a quake of the observed global scale; ii) arrival times are inconsistent with such a wave; iii) lighter-than-expected damage generally occurred expected in regions closest to deep water; iv) the sea level disturbance "hopped" across continents. (i and ii are the most important).

-Instead, we show that an alternate mechanism explains the observations; namely, atmospheric pressure fluctuations generated by the explosions, rather than land displacement, forces the ocean wave through a resonance mechanism occurring at the air-sea interface. We call this an acoustic-gravity wave because of this coupling between the air (acoustic) and ocean (gravity) waves.

-We demonstrate that all of the above observations are explained by this alternate hypothesis, which has ramifications for societally important factors such as risk assessment.

Though I am definitely not a tsunami expert, I am an ocean physicist very familiar with ocean and atmospheric waves and their dispersion relations. Still, it took me three reads to understand that this is their story. If I have it right, then I believe that this is a **great** story (though again I defer to the other tsunami-expert reviewers on the significance of this finding, which must have some history?) Personally, as I told Dr. White when I accepted this review, I was surprised as a non expert that the relatively small displacements from a volcano could generate such a global disturbance, but I had not thought of this mechanism. So, I believe that a well-written manuscript that clearly lays out this story in a physically intuitive way, and how the data are consistent with the new and inconsistent with the old mechanisms, would be well worth a prominent spot in Nature - maybe even a cover!

However, this is **not** that article as written. I believe that their results are reasonable, the mechanism plausible, and that they have the data to show what they are claiming. But for me at least, the article, which probably had too many typos for a Nature submission, was organized nearly

backwards from how I think most readers would want to learn the material, and in spite of lots of pretty plots I never saw the plots I actually wanted to see which were (among others) demonstration that the amplitude (i from the list above) and timing (ii) are consistent with air-wave and not water-wave speeds. Additionally, many of the figures were missing key legends and descriptors in the caption and so were hard to interpret. As a reader I was quite skeptical until the end, because I was not told what an acoustic-gravity wave was, and was being asked to trust the authors without being given an estimate that the expected response was the right order of magnitude or a quantitative demonstration that the travel times were right. So I suggest a reordering of the presentation and a major redo of the figures.

In terms of my observations-to-be-explained list in the first paragraph above, I think the authors did NOT show i), and they said but did not show ii), iii), iv). For example, take Figure 2. As a reader I want to be shown that the authors' claims that the observed speeds are inconsistent with wave-wave speeds and consistent with forcing by the air wave. But I'm asked to look at predicted arrival time numbers on a set of concentric rings in the top panel, then compare those to observed arrival times which must be read off a color map. Hardly a quantitative comparison. I'd want to see a plot of observed arrival time versus range, with error bars, together with the predictions for each theory on top. Same goes for the other claims which are that displacement is too small (i), damage was greater in regions without wide shelves (iii), and the continental hopping. Again, I'm not contesting these claims and in fact think they are plausible - but please show us, don't tell us!

In summary, I am supportive of this manuscript being published, but think a near total rewrite and redo of the figures is required.

Some select minor comments ensue; please note these are only a subset. I have attached my handwritten notes in hopes they will be useful in a revision.

-e.g. page 5: so much jargon. Surtseyan, cupressoid, tephra-finger, hydromagmatic, VEI are examples.

-Figure 1: no explanation that left is optical and right is infrared; nor what the upper and lowered images are which appear identical.

-Figure 2: no indication of meaning of colored bars in c; why must we compare colors (observed) and numbers (predicted). Please make a quantitative plot. 2b: I see no value to this plot.

-Figure 3: for me showed nothing useful towards the story above.

Figure E4: no legend.

P7: "common tsunamis:" there is nothing common about 7 km ocean depths!

P12: most readers are saying "finally they are explaining the mechanism!" It would greatly help the presentation if this or an abbreviated version of it were much earlier.

In this context, would a scaling argument showing the directly (DC) forced part, the so called inverse barometer effect, be useful in showing that that is *not* it but rather that the resonant (AC) coupling is important?

p20: Why is a range of speeds used? If this is used to demonstrate that agreement of predicted versus observed arrivals, tuning the former with the latter is problematic no? (I understand that the optimally computed speeds of sound are reasonable atmospheric values).

Referee #2 (Remarks to the Author):

Please see three attached PDFs

Dear colleagues,

The January 15, 2022 Tonga tsunami motivated and will keep motivating scientist works to understand this unusual world-wide tsunami. The contribution you submitted for publication to Nature is the best I had the opportunity to read these last weeks. I appreciated that you not only describe the concordance between atmospheric and ocean records, which was already very well described on social media, thanks to the incredible amount of data available. Your interdisciplinary team was able in a few weeks to adress the problem, trying to understand the physics behind this unusual tsunami and proposing a robust modelling approach. Congratulations for that.

I attach an annotated version of your manuscript where you will find wome minor comments + 2 additional references + two suggestions listed below :

1. The manuscript is silent about the local tsunami observed in the Tonga island. Indeed in the near-field, wave runups exceeding 15 m asl were observed (e.g. Kalokupolu, Tongatapu Island), thus suggesting that another source mechanism, most probably of volcanic or gravitational origin, generated waves several meters high on the coasts of the Tonga Islands, which is confirmed by eyewitness accounts, aerial photographs, and satellite images. A mention to this local tsunami, which is clearly different from the worldwide tsunami, should be added at the end of page 7 (end of section "Exceptional tsunami followed...").

2. I agree that there is a good agreement between the model and observations in terms of arrival times, but there are significant discripancies in terms of wave amplitude. This must be mentioned and discussed (e.g. a short sentence of explanation in the main text + a more detailed discussion in the methods).

I'm sure you will be able to address these comments.

With my best regards,
Raphaël Paris

Referee #3 (Remarks to the Author):

Key results: Explanation of the far reach and long duration of the far-field tsunami generated by the Hunga Tonga - Hunga Ha'apai eruption - really good agreement in timing and time series with open ocean DART buoys, analytical solution of the resonant interaction between the acoustic-gravity wave (AGW) and the surface gravity waves (SGW)

Originality: As the authors state, this is the first time that we have had data of this sort from an eruption of this magnitude and so understanding how the far field reacts is important and original. The analytical work synthesises earlier work (appropriately referenced) and extracts the parts that are pertinent in this case. This will be of great and immediate interest to people in my field as well as other adjacent fields.

Validity: Generally good - the analytical solutions derivation needs to be cleaned up before it can be published. It should be acknowledged that the pressure field doesn't match the nearfield pressure field but I don't think this should preclude publication because the paper is focused on the far-field and it is valid there.

Data and methodology: The data and methodology is generally good - there are some points in the minor comments below that need to be addressed but none of them are show stoppers

Clarity and context: Sometimes - especially in the methodology the language slips into almost 'bullet points' - they are no longer full sentences, e.g. Worth mentioning that... A thorough editing for language would significantly improve this paper. Also see comment re consistency of terminology below. The authors should emphasise more the fact that they are able to match DART buoy timeseries as well as just the arrival times. The abstract is clear and accessible as is the start of the paper. The conclusions are fine too but some of the interesting supplementary work is only mentioned in the summary and then only briefly - it would be good for the implications of this to be more considered.

Reference: previous literature cited appropriately

Inflammatory material: none

General comments:

Consistency of language. Authors refer to shock wave, acoustic-gravity waves, atmospheric waves, air-pressure jumps and similar terms - are these all the same thing? Are they subtly different? If so how? Ideally pick one term at the beginning and stick with it.

You have some great material in this but could do a better job of bringing it all together in the end and explaining the implications to what was seen at HTHH

Minor comments:

line 39 It is probably more to do with the area over which the displacement occurs rather than the

total volume that determines the ability to become a trans Pacific tsunami

line 44 Unclear what you are saying is the necessary but not sufficient condition. Is this just the size of the explosive pressure? Is it the acoustic gravity waves? What would be sufficient? This does not make that clear

line 49/84 and elsewhere Capitalise Plinian/Surtseyan etc

line 50 vulcanian? maybe volcanic

line 57 - refs to Pinatubo and Taupo - no refs for Krakatau or Kamchatka

line 80 query does the edifice just refer to the small cone that formed between Hunga Tonga and Hunga Ha'apai? The underlying sub aerial part has been there for far longer

line 84 Hunga volcano has produced...

line 90 intercalated - is this a technical term? it certainly isn't a common term.

line 96-98 centre of the new cone had already been destroyed before the Jan 15 eruption (probably the Jan 14 eruption as satellite image pre eruption shows the middle missing)

line 104 remove simultaneously

line 105 metric-sized is meaningless - do you mean tsunami waves on the order of a metre high? (Amplitude or peak to trough?)

line 121 common tsunamis - better to just say earthquake triggered tsunamis. Could also just say the shallow water wave limit as it has nothing to do with the wave generation but means that they had to be forced waves

line 126-129 - this seems cumbersome - can you not just indicate the shallow water wave speed

line 135 figure only shows 1.5-2 days

line 136 'resulting wave amplitudes over the oceans' this needs to be rephrased. Also worth noting that sea level gauges only tell some of the story and run-up heights in some locations in Tonga were significantly higher.

Fig 2a - Please colour code contours with the same scale as the dots - this will make it far easier to see the differences

Fig 2b in caption refer to height but amplitude in figure - make sure it is clear what is being plotted.

Fig 2c Sea level records shown here do not match with heights given in Fig 2a - obviously taken from other locations. This doesn't invalidate what you are saying but needs to be explained. Also, are the coloured bands calculated from an external model or are these just highlighting where the observed disturbances occur?

Fig 2d Underlying picture is brightness temperature which is not explained in the caption. The caption says that this shows good agreement with actual barometric pressure anomalies but how this is shown by this figure is not obvious (if this is where the bands in Fig 2c are calculated from then state that.

line 153 remove 's

line 159 It shows they are correlated, I am not sure it shows they are intrinsically linked

line 163 extended data fig 3 does not show extra atmospheric waves - these can be seen in the animations.

line 168 Is this really coupled? From what you have said it sounds like you reproduce the atmospheric disturbance by assuming it spreads radially around the earth and reduces in height. This is used to force the tsunami model. I don't consider that coupled.

line 173 'our numerical modelling of a tsunami forced by ...' Is this the same thing as the coupled model you mention above (it is a far more accurate description of it but the way you say it sounds like it is something different

line 176-177 I understand what you are saying but I think it would be better to state that it can be rapidly generated or something like that which make it appear to "jump"

line 179 further excited

180-183 Point to details in methodology?

line 183 You miss the most important part of this - that your model actually reproduces the time series well at many locations

185 What do you mean by overall maximum uniform wave heights distribution

line 191 Fig 3 incorrect terminology - its not the ocean swell that it is resonating with it is the tsunami waves

195-196 I can't see evidence that the maximum wave heights are in agreement with observations - why don't you plot the DART buoys on 3c in the same colour scheme as the modelling - this would show whether that claim is true or not.

line 198 What is the novel model? meteotsunamis are already known about. Far field wave from Krakatau have been shown to be from this mechanism - you even quote studies that look at Taupo etc in this light.

line 204-206 What is your atmospheric model based on? Are you actually modelling this or just giving an explanation? I really don't follow what you mean by 'distribution of sources over the volcanic column' it sounds like you are talking vertically but then you are saying this is the air-sea interface. Isn't it more to do with refraction and wave speed at different heights? I am not sure how diffraction comes into it

line 207 At these scales 'unperturbed' is irrelevant

lines 207-209 The point is that it is a forced wave. High pressure oscillation doesn't really mean anything

line 212 second tsunami wave train - make sure you are clear what you mean by this - earlier you were attributing this to multiple passes of the AGW, again sloppy use of multiple terms. When you say existing tsunamis do you mean the tsunami wave train first generated by the AGW or do you mean the original local tsunami

line 217-218 Incorrect terminology, it is not the long swell - swell refers to deep water waves - these are shallow or intermediate water waves. Also this whole part suggests that this is a different mechanism to the previous paragraph (212-216) I don't see how this is different.

line 219-221 It is not just significantly inhibited - for resonance to occur the speeds need to be at least in the same ballpark and ideally the same speed... You are just repeating yourself here.

line 222-223 Worth explaining why you do get tsunami waves in Caribbean/Mediterranean - is it only in the deeper places of these ocean basins?

lines 229-230 Pacific Islands (oceanic is a tautology here)

line 231-232 or seismic tsunamis which are linear sources and are the most common sort of tsunami (point sources tsunamis are not that common)

233-234 It is more that they have energy being pumped in to increase their size. Also the atmospheric forcing has managed to excite relatively linear shallow water waves which do travel with minimal energy loss...

line 233 also that enough energy gets pumped in that they can reach the size of seismic tsunamis

Fig 4 This fig suggests that 1, 2 and 3 are successive moments in time but in 2 there is already a tsunami some of which is reaching beyond the AGW. The first wave will be a bound wave pushed by the AGW and there will be a wave train behind it - later waves may be reinforced by later AGW

line 242 not ocean swells

line 246 analytical analysis (is there any other kind?)

line 256-257 What evidence do you have that they are more common than previously thought? It seems like you might get a big one every hundred years or so (last big one being Krakatau)

lines 339-340 'constraint its origin' 'oceanic data concern sea-level accurate records' - these do not make sense

342 complemented? It would take a lot more to complete

344 concerns - do you mean consists of?

346 specifics

352-353 It is not common to talk about data not being harmed - affected would be a better word.

What do you mean by data recalculated to millimetres. In fact right through to 356 does not really make sense and needs to be rewritten

366 peak to peak wave height? I assume you mean peak to trough (or vice versa) you are not going to get the maximum wave height by measuring peak to peak?

367-369 This seems like assigning numbers for the sake of seeming rigorous without adding anything - You could just say 'we only kept arrival times where there were only minimal doubts about the arrival (or something like that)

374

Also this whole section shows that you don't have a coupled model - you empirically estimate the pressure forcing (which is completely off in the nearfield as can be seen by looking at the BoM data from Nuku'alofa but does a good job in the far field)

line 431 - limited rather than constrained

Analytical solution

This is some interesting analysis but this entire section is very sloppy and not consistent in the terms used - so while the overall analysis could be valid it is poorly presented and so difficult to ensure it is correct. This is a shame because it could be very interesting. Especially if it were better connected with the implications for HTHH. How well does this analytical model fit what was seen/modeled in the open ocean?

Some specific issues are given below

Zeta appears to be the free surface in (7) but you've just said it is h . You haven't defined in this what you consider the free surface (the top of the atmosphere or the ocean surface)

453-455 First you assume that the interface is rigid and then in (6) $z = -1/\mu$ which is below where you are considering (assuming z positive up - not necessarily a given here...)

455-457 τ is half the duration in your example

459 the equations state $z = \zeta$ but you say they are both on $z = 0$ - these can't both be true. In the text you talk about $z = h$... ζ is never mentioned in the text but I assume this is the free surface - this seems to just be copied from ref 32 as is using the same terms.

Equation 10 cannot be obtained by expanding equation 7

489 typo

496 - inconsistent with reference paper

ED Fig5 A better explanation of what is being shown would be useful. The top seems to be the Gaussian pressure disturbance (and I can just make out A on the vertical axis) but is that the SGW solution also plotted on it? I think it is useful both being plotted on the same graph and then the SGW being plotted again magnified below but this needs to be stated clearly. Also these are solutions for radius r but you are plotting positive and negative r ... Is there any physical significance

to the negative solutions? If not these should not be plotted. You identify earlier tsunami amplitude amplified due to AGW but this appears to be only in the negative r domain so may be unphysical.

Author Rebuttals to Initial Comments:

RESPONSE TO COMMENTS FROM THE REVIEWERS

Referee #1

Please see attached PDF

1. I summarize what I understand to be the results of this paper here:

-Tsunamis are usually thought to be forced by the direct displacement from land (e.g. new lava, earthquakes and landslides) to ocean. In the recent Tonga quake, we show that this mechanism is inconsistent with the observations (which for the first time are global in scale), in several ways: i) displaced land volume is too small to generate a quake of the observed global scale; ii) arrival times are inconsistent with such a wave; iii) lighter-than-expected damage generally occurred expected in regions closest to deep water; iv) the sea level disturbance “hopped” across continents. (i and ii are the most important).

-Instead, we show that an alternate mechanism explains the observations; namely, atmospheric pressure fluctuations generated by the explosions, rather than land displacement, forces the ocean wave through a resonance mechanism occurring at the air-sea interface. We call this an acoustic-gravity wave because of this coupling between the air (acoustic) and ocean (gravity) waves.

-We demonstrate that all of the above observations are explained by this alternate hypothesis, which has ramifications for societally important factors such as risk assessment.

Though I am definitely not a tsunami expert, I am an ocean physicist very familiar with ocean and atmospheric waves and their dispersion relations. Still, it took me three reads to understand that this is their story. If I have it right, then I believe that this is a *great* story (though again I defer to the other tsunami-expert reviewers on the significance of this finding, which must have some history?) Personally, as I told Dr. White when I accepted this review, I was surprised as a non expert that the relatively small displacements from a volcano could generate such a global disturbance, but I had not thought of this mechanism. So, I believe that a well-written manuscript that clearly lays out this story in a physically intuitive way, and how the data are consistent with the new and inconsistent with the old mechanisms, would be well worth a prominent spot in Nature – maybe even a cover!

R1: We are delighted to read that the reviewer is very positive about the merit of our work and thinks that the work – if improved – would potentially be worth a prominent spot in Nature. We also completely take the point of the reviewer that the text needed considerable work to make it more fluid and easier to follow, and to better appeal to the broad readership of Nature. Following the criticisms by the reviewer, we worked to improve this aspect of the paper, as detailed below.

2. However, this is **not** that article as written. I believe that their results are reasonable, the mechanism plausible, and that they have the data to show what they are claiming. But for me at least, the article, which probably had too many typos for a Nature submission, was organized nearly backwards from how I think most readers would want to learn the material, and in spite of lots of pretty plots I never saw the plots I actually wanted to see which were (among others) demonstration that the amplitude (I from the list above) and timing (ii) are consistent with air-wave and not water-wave speeds. Additionally, many of the figures were missing key legends and descriptors in the caption and so were hard to interpret. As a reader I was quite skeptical until the end, because I was not told what an acoustic-gravity wave was, and was being asked to trust the authors without being given an estimate that the expected response was the right order of magnitude or a quantitative demonstration that the travel times were right. So I suggest a reordering of the presentation and a major redo of the figures.

R2: Again, we completely take the point of the reviewer that our submission needed a major shakeup in terms of flow, clarity, pithiness, and comprehensibility by a non-expert audience, and many of the suggestions/criticisms by the reviewer indeed point us in the right direction, which we much appreciate. The reviewer, however, suggests a major restructuring of the paper, given that he/she considers that the manuscript is "organized nearly backwards from how he/she thinks most readers would want to learn the material". To be clear, we made a considerable effort to re-write the paper to address these shortcomings, but we did not completely implement this "reversed" approach suggested by the reviewer, as we feel that the structure we outlined in our initial submission – notwithstanding the need for substantial improvements of the text – is more appropriate, as we shall explain. Our reasoning concerns the fact that indeed the air-water coupling triggered by volcanic explosions has been proposed, in the past, as a possible or feasible generation mechanism the generation of global tsunamis, as has been hypothesised for those created by the Taupo and Krakatau eruptions (the "history" that the reviewer suggests that probably exists but of which he/she is not aware, as a non-expert). The novelty of our study, however, lies in demonstrating, for the first time, that this theoretical concept is not only correct, but it can now be tested, corroborated, and adequately explained – numerically and analytically – on account of the recent Tonga eruption and tsunami and the existing worldwide wealth of available data. In fact, this is particularly well illustrated by a comment of reviewer #2, which states "*I appreciated that you not only describe the concordance between atmospheric and ocean records, which was already very well described on social media, thanks to the incredible amount of data available. Your interdisciplinary team was able in a few weeks to address the problem, trying to understand the physics behind this unusual tsunami and proposing a robust modelling approach*". In this sense, we feel that our existing structure is suited to convey this message. We therefore opted to undertake significant improvements of

the text in line with the reviewers' comments but decided to keep the general structure we laid out in our initial submission. In detail, we have significantly re-written the abstract, introduction and the subsection about the eruption to introduce key concepts earlier on, and make the text more fluid, pithy and leaner, and also to make the text comprehensible by a non-expert audience (i.e. by purging all unnecessary jargon in the revised version). We also improved the figures and added new ones to demonstrate that the observed tsunami metrics are consistent with our proposed mechanism and not with a point-source mechanism, exactly as the reviewer suggests.

3. In terms of my observations-to-be-explained list in the first paragraph above, I think the authors did NOT show i), and they said but did not show ii), iii), iv). For example, take Figure 2. As a reader I want to be shown that the authors' claims that the observed speeds are inconsistent with wave-wave speeds and consistent with forcing by the air wave. But I'm asked to look at predicted arrival time numbers on a set of concentric rings in the top panel, then compare those to observed arrival times which must be read off a color map. Hardly a quantitative comparison. I'd want to see a plot of observed arrival time versus range, with error bars, together with the predictions for each theory on top. Same goes for the other claims which are that displacement is too small (i), damage was greater in regions without wide shelves (iii), and the continental hopping. Again, I'm not contesting these claims and in fact think they are plausible - but please show us, don't tell us!

R3: We completely take the point of the reviewer about quantitative rigour, and the need to improve the figures to better demonstrate our arguments and the physical evidence to support our models.

Accordingly, we colour coded the point-source tsunami travel time map with the same scale as the observations (dots) to make any quantitative comparison easy (see new Fig. 2a). We also added: 1) a plot, as suggested by the reviewer, showing observed arrival time versus distance, with error bars, together with the predictions for both point-source and air-driven tsunamis (see new Extended data Fig.1), and 2) a map (Fig. 3b). As the reviewer predicted, this plot illustrates very clearly that the real tsunami travel times are correlative with a moving atmospheric source and not with a point-sourced tsunami. The same applies to our claim on the minimal dissipation of the Tonga tsunami energy, which we clarified by re-working the Fig. 2b and adding a new figure (Extended data Fig. 2) to quantitatively show the rapid dissipation of point-source tsunami in comparison with observations.

4. In summary, I am supportive of this manuscript being published, but think a near total rewrite and redo of the figures is required.

R4: As suggested by the reviewer, we have significantly re-written the manuscript, partially re-structured it (to tell our "story" at the beginning), improved its presentation, flow, comprehensibility by a non-expert audience, and qualitative rigour. Figures were also re-worked (Figs. 1, 2, and 3) and new ones were added (Extended Data Figs. 1 and 2) to quantitatively support our work hypothesis.

5. Some select minor comments ensue; please note these are only a subset. I have attached my hand written notes in hopes they will be useful in a revision.

R5: We are really disappointed that the PDF quality prevented us from reading many of the comments by the reviewer. We addressed below all those we understood, but unfortunately, we did not understand the meaning of several of these comments, as we were unsure what the reviewer wrote.

6. -e.g. page 5: so much jargon. Surtseyan, cupressoid, tephra-finger, hydromagmatic, VEI are examples.

R6: We have purged the manuscript of all highly niche jargon, including all the words listed by the reviewer. These were either removed altogether, or substituted by simple expressions comprehensive to a general audience (e.g. substituting “surtseyan” by “shallow-water explosions”).

7. Figure 1: no explanation that left is optical and right is infrared; nor what the upper and lowered images are which appear identical.

R7: The reviewer is right, and we corrected this. We have also modified the display/arrangement of the subfigures to one that is more intuitive to read.

8. Figure 2: no indication of meaning of colored bars in c; why must we compare colors (observed) and numbers (predicted). Please make a quantitative plot. 2b: I see no value to this plot.

R8: We implemented all the reviewer’s suggestions on Fig. 2. Fig. 2a is now colour coded to easily compare, using the same scale, point-source with observed tsunami travel times. Fig. 2b has now been improved to show the comparison of predicted point-source (also supported by the numerical simulation in Extended Data Fig.2) with recorded maximum tsunami wave amplitudes, and how the latter do not follow the trend of typical point-sourced tsunamis, which is characterized by an exponential decay as a function of increasing distance to the source. In Fig 3c, we have now highlighted more clearly the meaning of the colour bars.

9. Figure 3: for me showed nothing useful towards the story above.

R9: Figure 3 is now completely re-worked to support the manuscript narrative. New Fig 3a is added to demonstrate that atmospheric-driven tsunami travel time is consistent with observations (also supported by the new Extended Data Fig. 1). Fig. 3b is re-worked

to show that our numerical model is consistent with: 1) a fast-travelling tsunami, as it produces waves that “jump” from an ocean to another, 2) a global reach tsunami, as the acoustic-gravity waves are able to excite the ocean-surface and generate the tsunami as they propagate across the globe, 3) a long-duration tsunami, as the acoustic-gravity waves are able to generate a new tsunami in its way back to the source area. Fig 3c also illustrates how wave amplitude increases through resonance when acoustic-gravity waves travel over deep-water oceanic trench. Fig. 3d shows that our numerical model is consistent with the global reach of the Tonga tsunami and that it is capable of reproducing the overall maximum wave amplitudes observed at many open-ocean locations.

10. Figure E4: no legend

R10: Extended Data Fig.4 (now Extended Data Fig.6) is re-worked and the legend is added accordingly.

11. P7: “common tsunamis:” there is nothing common about 7 km ocean depths!

R11: We rephrased this to avoid any unclarity and the sentence now reads: “*Our analysis suggests that the Tonga tsunami propagated at a speed of ca. 1000 km/h, showing no significant decrease from deep to shallow water, as expected for earthquake-triggered and point-sourced tsunami waves as their phase speed (c) is linearly dependent of the water depth (h) over which they propagate (i.e., $c = \sqrt{gh}$, where g is the gravity acceleration).*” (See pages 10-11 of the annotated MS)

12. P12: most readers are saying “finally they are explaining the mechanism!” It would greatly help the presentation if this or an abbreviated version of it were much earlier.

R12: We have added earlier in the text introductory sentences on our explanation of the mechanism behind the unusual aspects of the observed Tonga tsunami but without a complete re-structuring of the manuscript as explained in our response to the reviewer comment #1. They now read: “*We explain this unusual tsunami by a moving atmospheric source mechanism that forces the ocean surface and accompanies both the formation and propagation of tsunami waves. Crucially, the profusion of atmospheric and sea-level readings that recorded this event across the globe and the availability of high-resolution geostationary satellite observations of the effects of the source and propagating acoustic-gravity waves provide a unique opportunity to finally unlock the most enigmatic aspects of air-water coupling and tsunami generation via ocean forcing and resonance. The Tonga tsunami, hence, represents the first opportunity to constrain the actual range of physical model parameters for the formation of global tsunamis by acoustic-gravity waves, allowing us to move beyond a “proof of principle” into the development of useful forecasting tools for disaster risk reduction.*” (see page 5 of the annotated MS)

13. In this context, would a scaling argument showing the directly (DC) forced part, the so called inverse barometer effect, be useful in showing that that is *not* it but rather that the resonant (AC) coupling is important?

R13: We understand the concern of the reviewer in showing whether the inverse barometer effect or the resonant coupling is important. However, in the case of Tonga tsunami we believe that both are important, as the first (i.e. DC) provides relevant information on the arrival time of the tsunami while the second (i.e. AC) helps explaining the high tsunami recorded at coasts close to oceanic areas favouring the air-water resonance. In general, inverse barometer effect leads to relatively small wave amplitudes (~1 cm height for 1 hPa pressure jump) that can be amplified via resonance. The latter effect is strongly dependent of the water depth over which the pressure wave propagates and ,therefore, requires high-resolution bathymetric data to be accurately estimated. Such an assessment is beyond the scope of our study as we provide an explanation of the event at a global scale using coarse bathymetric data.

14. p20: Why is a range of speeds used? If this is used to demonstrate that agreement of predicted versus observed arrivals, tuning the former with the latter is problematic no? (I understand that the optimally computed speeds of sound are reasonable atmospheric values).

R14: Yes, the speed range is used to estimate the optimal speed that allows reproducing the observed arrivals with the minimum error. We edited the text to improve the clarity: *“From these comparisons, we obtain the optimal speed value of 322 m/s, as it allows the best agreement in pressure jump arrivals between the simulations and observations (error less than 0.042, see Extended Data Fig. 5b).”* (See page 31 of the annotated MS)

15. Hand-written comments:

L20-22: “Violent volcanic explosions, however, could potentially cause global tsunamis^{1,5} by travelling acoustic-gravity waves^{6,7,8} that excite the atmosphere-ocean interface”: From above?

R: Yes. We believe this is implicit and did not make any change to the text.

L25: “.... *atmospheric -driven tsunamis*”: Not shown yet.

R: The abstract was re-written to address this problem.

L26-27: “...*from the entire globe*”: really?

R: Changed to “*from across the globe*” (see page 2 of the annotated MS).

L29-30: “Direct correlation between the tsunami and the acoustic-gravity waves’ arrival times confirms that the generation and propagation of these phenomena are closely linked.”: ?

R: We are unsure what the reviewer means here. Given that we are able to demonstrate this statement throughout the manuscript we kept it as it is.

L31: “unusual”: unusually

R: Corrected.

L31: “is”: are

R: Corrected.

L33-34: “ *differential global tsunami height distribution* ” :?

R: Phrase changed to avoid unclarity. It read now: “*This coupling mechanism has clear hazard implications, as it explains the unexpected global tsunami height distribution, characterized by higher waves along landmasses that rise abruptly from long stretches of deep ocean waters.*” (See page 2 of the annotated MS).

L45: “..that the explosive pressure is ...”

R: Changed accordingly.

L46: “..progressive atmospheric acoustic-gravity ...”

R: Changed accordingly.

L47: “Acoustic-gravity waves are...”: Ref

R: Reference added accordingly.

L49-50: “..plinian and sub-plinianvolcanian...”: Jargon

R: All jargons were removed.

L52: “....and thus can conversely travel significant distances...”: ? Does not follow.

R: “conversely” removed from the sentence.

L55: Reference 13 “circled”

R: We do not understand what the reviewer means with this, but we revised all references and corrected any possible errors in referencing and referencing numbers.

L57: “... with as few as three known cases^{7,8}”: ?

R: This section was completely re-written and this specific phrase now reads: *However, the generation of transoceanic volcanic-tsunamis of global impact, although rare, is theoretically possible through air-water coupling⁹, as possibly illustrated by the ca. AD 200 Taupo (New Zealand)⁸, the 1883 Krakatau (Indonesia)^{1,5,10,11,12}, and the 1956 Bezymianny (Kamchatka)¹³ eruptions and associated far-reaching tsunamis.* (See page 3 of the annotated MS).

L63: “...with the ocean swell...”: not mentioned previously.

R: This was corrected in the text.

L64: “As a result of this,”: of what?

R: Changed to avoid unclarity.

L65-66: “*the hazard extent of atmospheric-driven tsunamis triggered by volcanic activity (also known as volcano meteorological tsunamis^{4,7}) is still largely unknown and rarely considered.*”: Isn't this what is proposed? If so define and introduce earlier.

R: Indeed, and this was introduced earlier in our revised introduction.

L68: “.....enigmatic atmospheric waves.....”: ?

R: changed to “noticeable” – however, the term “enigmatic atmospheric waves” is in the original reference provided.

L69: “*provides the perfect opportunity*”: provide an opportunity.

R: Changed to “unique opportunity”

L70: “...their intervening mechanisms.”: ?

R: Expression removed.

L77-78: “The 15th January 2022 volcanic explosion ~~that recently took place~~ at the Hunga Tonga-Hunga Ha'apai volcano is ~~perhaps~~ one of the largest in the last 30 years”

R: Suggestions implemented.

L79: “...of hydromagmatic eruptive style..”: Jargon

R: Jargon removed

L89,94,98: “..surtseyan cupressoid tephra-fingerhydromagmatic...Volcanic Explosivity Index...”: Jargon

R: Jargon removed

L94: “on ~~the~~ January 15th”

R: Corrected

L99-100:” *the highest eruptive columns of the satellite age*”: why is this relevant?

R: This is relevant because one of the key parameters used to measure the size of an explosive eruption, i.e. the explosivity of an eruption, is the height of the eruption column. As such, the fact that the eruption column was one of the highest ever measured since modern satellite readings started, is very illustrative of how violent this eruption was. We therefore opted to keep this statement.

L102: “*unusual pattern of concentric atmospheric acoustic-gravity waves it created*”

R: we are unsure what the reviewer meant when highlighting this sentence... but we decided to change “*concentric*” to “*concentrically-propagating*” hoping that this can clarify the meaning.

L105: “by metric-sized tsunami”: by meter-sized tsunami

R: We corrected this by being more specific in our meaning, i.e. by introducing actual values.

L117,118,121,122:” exceptional propagation speed, unprecedented duration, faster speed, much earlier” : Quantify

R: Indeed, we modified the text accordingly to improve quantitative rigour. This was done by introducing specific values in the following sentences, using examples extracted from our dataset and with a clear reference to the revised Figures 2 and 3, as well as the supplementary material.

L126-129: “ *The tsunami propagated from the volcano area with an almost uniform speed of ca. 1000 km/h, which is comparable to common tsunamis propagating in deep water (over 7 km depth) but about six times higher than their speed in shallow water (ca. 200 m depth).*”: 7 km is not common

R: We took the point of the reviewer and modified the text accordingly (see response to comment#11).

L143: “simulated”: Standard

R: Changed.

L151: “volcanogenic”: ?

R: Changed to “*volcanic*”.

L53: “*minimal transoceanic wave’s dissipation*”: Not quantified.

R: We took the point of the reviewer and produce new simulations of point-source tsunami (Extended data Fig.2) then compare the simulated wave amplitudes with data (Fig. 2b). We also edited the text and introduced values to illustrate the statement.

L155-157: “*A comparison of the atmospheric and oceanic data shows a direct correlation between the first passage of the air-pressure jump and the onset of the tsunami in many locations around the globe (Fig. 2c).*”: Not and does not

R: We are unsure what the reviewer meant here, but we slightly edited the text to improve clarity.

L161-162: “... supporting the atmospheric (acoustic-gravity wave) origin of the Hunga Tonga-Hunga Ha'apai tsunami.”: No

R: Sentence reworded to improve clarity.

L163-165: Apparently the orange + yellow bands. Put in caption.

R: Indeed, thanks for spotting that, explanation of the bands is now inserted in the caption of Fig. 2.

L168:"A"

R: Corrected.

Page 10: Why not a side by side comparison of obs vs. – Standard , air-sea coupling models

Focusing on time of Arrival and amplitude

Show your model is better! Don't tell us

R: We took the point of the reviewer, produced the graphs suggested for tsunami travel time of observations versus point-source and atmospheric tsunami models (Extended Data Fig. 1). We also produced simulations of point-source tsunami wave heights and compare them with observations (Fig. 2b). Moreover, throughout the text we improved quantitative rigour.

Page 11: Would be so much clearer if this came first!

R: We took the point of the reviewer and We have added earlier in the text introductory sentences on our explanation of the mechanism behind the unusual aspects of the observed Tonga tsunami (see response to comment#12).

L202: are set by

R: Corrected.

L204-205: "the explosion can be pictured as a distribution of sources over the volcanic column"

R: Rephrased to avoid unclarity. It now reads: "*At the air-water interface, the explosion can be pictured as a flow through a circular opening or a distribution of sources over the volcanic column cross-section.*" (See page 20 of the annotated MS).

L209: inverse barometer effect?

R: Indeed, thanks for spotting that, explanation on the “inverse barometer effect” added accordingly.

L 216-219: but you just said atmos>>ocean speed

R: Text reworked to improve clarity and edited accordingly “*This explains an earlier arrival of the tsunami across the globe with respect to a point-sourced tsunami (Extended Data Fig. 1), as the acoustic-gravity waves travel much faster than the ocean-gravity waves in intermediate to shallow ocean water depths.*” (see page 20 of the annotated MS). “*Tsunami can also amplify under Proudman resonance³¹, when acoustic-gravity wave speed matches the speed of the ocean long-waves in very deep water*” (see page 21 of the annotated MS)

L 218: then what?

R: Text edited to address the reviewer question: “*Tsunami can also amplify under Proudman resonance³¹, when acoustic-gravity wave speed matches the speed of the ocean long-waves in very deep water, leading to higher-amplitude waves (Fig. 3c) that propagate under gravity towards the coast, i.e. second tsunami wave train of later arrival (Fig. 3b),*” (see page 21 of the annotated MS)

L 219: number them

R: Number inserted.

L 231: remove “contrastingly”

R: Removed.

L234: add “more”

R: Added.

L334: change “concern” to “comprise”

R: Changed.

L381: Typo

R: Corrected.

L420 add “ref.”

R: Added.

L423: remove “the”

R: Removed.

L428: “two times” to “twice”

R: Corrected.

L430: It is

R: Rephrased.

L436: All analysis is analytical?

R: Changed to “Analytical model”.

Pages 22-23 comments

R: We have corrected the typos and revised this section entirely. To avoid inconsistency in the terms, each of the two presented models has now its own boundary conditions. Note that we only present the minimum number of equations required to describe the model and the results applied.

Referee #2

Please see three attached PDFs

Dear colleagues,

The January 15, 2022 Tonga tsunami motivated and will keep motivating scientist works to understand this unusual world-wide tsunami. The contribution you submitted for publication to Nature is the best I had the opportunity to read these last weeks. I appreciated that you not only describe the concordance between atmospheric and ocean records, which was already very well described on social media, thanks to the incredible amount of data available. Your interdisciplinary team was able in a few weeks to address the problem, trying to understand the physics behind this unusual tsunami and proposing a robust modelling approach. Congratulations for that.

We sincerely thank the reviewer for his very encouraging feedback, and we are delighted to hear that he considers our approach one of the best he reviewed on the subject. We are also absolutely delighted to hear that the reviewer – a renown international tsunami expert – immediately understood the novelty of our approach, which goes well beyond the observation that the Tonga tsunami was “volcano-meteorological” tsunami to propose a robust physical explanation of the phenomena and a feasible numerical model capable of reproducing the observed data. We also thank the reviewer for his comments, which greatly helped to improve the manuscript.

I attach an annotated version of your manuscript where you will find some minor comments + 2 additional references + two suggestions listed below:

1. The manuscript is silent about the local tsunami observed in the Tonga island. Indeed in the near-field, wave runups exceeding 15 m asl were observed (e.g. Kalokupolu, Tongatapu Island), thus suggesting that another source mechanism, most probably of volcanic or gravitational origin, generated waves several meters high on the coasts of the Tonga Islands, which is confirmed by eyewitness accounts, aerial photographs, and satellite images. A mention to this local tsunami, which is clearly different from the worldwide tsunami, should be added at the end of page 7 (end of section "Exceptional tsunami followed...").

R1: We agree with the reviewer comment and accordingly added his suggestion within the Section on the “Exceptional tsunami followed the Tonga-Hunga Ha’apai volcanic explosion” once we mentioned the tsunami wave record at Tonga Island: “*At a distance of ca. 67 km, the tide gauge of Tonga Island recorded a maximum wave amplitude of*”

1.14 m (Fig. 2b, Supplementary Information Table 1) that, however, does not reflect the full near-field tsunami impact as high runup heights (ca. 15 m) were also observed in Tongatapu Island, Tonga Archipelago, suggesting the contribution from a local tsunami source (likely of volcanic or gravitational origin).” (See page 11 of the annotated MS).

2. I agree that there is a good agreement between the model and observations in terms of arrival times, but there are significant discrepancies in terms of wave amplitude. This must be mentioned and discussed (e.g. a short sentence of explanation in the main text + a more detailed discussion in the methods).

R2: As suggested by the reviewer, we briefly mentioned the model vs. data time series comparison in the main text and added a more detailed discussion in the method section.

In the main text: *”It closely reproduces, at many open-ocean locations, both the observed maximum wave amplitudes (Fig. 3d) and the recorded time series (Extended Data Fig. 6) but with certain discrepancies (Methods)” (see pages 16-17 of the annotated MS)*

In the Methods: *” Overall, the numerical simulation results fairly reproduce the signals recorded at the open-sea DART stations, particularly in what concerns the tsunami arrival time and the first wave (Extended Data Fig. 6). They, however, show some discrepancies in terms of the late-arrival wave amplitudes. We believe that the quality (i.e., horizontal resolution) of the bathymetric data used here strongly affects the modelling results, leading to such inconsistency between the simulated and recorded waveforms. Another aspect that is not covered here and might also influenced the modelling results is the dispersive behaviour of the waves propagating away from the source. Moreover, some sea-level records might also include tsunami waves that are possibly triggered by a secondary point-source of volcanic or gravitational origin.” (See pages 33-34 of the annotated MS).*

I'm sure you will be able to address these comments.

With my best regards,

Raphaël Paris

minor comments:

3. Page 3: : For the 1883 Krakatau far-field tsunami, please cite the pioneering work of Harkrider & Press (1967 – see file attached) and Pelinovsky et al. (2005 – see file attached).

R3: References added for the 1883 Krakatau tsunami as suggested by the reviewer.

4. Page 4: Check the order of references.

R4: References were checked, and corrected numbers inserted.

5. Page 4: Is this really needed to state that the model used is the latest advance? What is the latest advance? It is always questionable.

R5: We agree with the reviewer's concern and revised the text accordingly, by completely removing this statement.

6. Page 5: Many authors describe these waves as "Lamb waves". Is it also your opinion? if so it would be nice to mention this briefly.

R6: In the general sense, in acoustic-gravity wave (AGW) theory, we refer to longitudinal compression-type waves propagating under the effects of gravity, in any continuous medium, such as gas, liquid, or solid. Thus, we look at AGWs from a broad perspective, which under specific boundary conditions have specific names, and Lamb waves would be one of these but not in the specific example given in the paper. For example, AGW travelling in the water layer are known as hydro-acoustic waves; on the other hand, considering an elastic sea bottom, under certain conditions AGWs can couple with the solid (elastic) boundary and travel as Scholte waves, and in the absence of the water layer, say at the shoreline, they become Rayleigh waves or Lamb waves under certain elastic conditions. Including elasticity is extremely interesting when analysing AGWs travelling in water/solid but here its contribution will be negligible. We have revised the description of AGWs to give a wider perspective: *"Acoustic-gravity waves are low frequency compression-type sound waves propagating under the effects of gravity at high speed close to that of sound of the medium (e.g., ca. 340 m/s in air and 1500 m/s in water)¹⁶. As such, they can reach significant distances away from the source before dissipating¹⁴. Acoustic-gravity waves propagate into different media, becoming hydro-acoustic (in water layer), Scholte (in elastic sea-bed) or Rayleigh-Lamb (in elastic layers)^{17,18}"* (See pages 3-4 of the annotated MS).

7. Page 9: In the far-field. In the near-field the ocean wave patterns observed on tide gage records mostly correspond to the local tsunami (i.e. generated by the explosion + landslide or pyroclastic flow).

R7: We agree with the reviewer comment and revised the text accordingly.

8. Page 12: The eruptive style remains unknown, although I agree that it was probably phreatoplinian. At this stage, you may write “violent hydromagmatic explosion”.

R8: We changed to “violent volcanic explosions” to be comprehensible by a non-expert audience.

9. Page 13: Recall figure 3b here

R9: Done.

10. Page 13: Except in the Tonga islands, where wave runup locally exceeded 15 m asl.

R10: *Rephrased accordingly to avoid any unclarity. The sentence now reads: “It is thus explained why t significant tsunami amplitudes were recorded at distant coasts like Japan (ca. 1 m) and along the western margin of South America (ca. 1.43 m in Chile) (Fig. 2b, Supplementary Information Table 1).”* (See page 22 of the annotated MS).

11. Page 13: Call figure 2a and suppl Table 1 (Excel file).

R11: Done.

12. Page 14: Figure 4

R12: Done.

Referee #3

Key results: Explanation of the far reach and long duration of the far-field tsunami generated by the Hunga Tonga - Hunga Ha'apai eruption - really good agreement in timing and time series with open ocean DART buoys, analytical solution of the resonant interaction between the acoustic-gravity wave (AGW) and the surface gravity waves (SGW).

Originality: As the authors state, this is the first time that we have had data of this sort from an eruption of this magnitude and so understanding how the far field reacts is important and original. The analytical work synthesises earlier work (appropriately referenced) and extracts the parts that are pertinent in this case. This will be of great and immediate interest to people in my field as well as other adjacent fields.

We sincerely thank the reviewer for her feedback, and once again we are delighted to hear that the reviewer considers our work robust, novel, and of potential wide appeal to Nature readership.

Validity: Generally good - the analytical solutions derivation needs to be cleaned up before it can be published. It should be acknowledged that the pressure field doesn't match the nearfield pressure field but I don't think this should preclude publication because the paper is focused on the far-field and it is valid there.

We are also glad that the reviewer considers the work robust, notwithstanding aspects that need improving and that indeed we worked to improve, following the suggestions by the reviewer.

Data and methodology: The data and methodology is generally good - there are some points in the minor comments below that need to be addressed but none of them are show stoppers.

Clarity and context: Sometimes - especially in the methodology the language slips into almost 'bullet points' - they are no longer full sentences, e.g. Worth mentioning that... A thorough editing for language would significantly improve this paper. Also see comment re consistency of terminology below. The authors should emphasise more the fact that they are able to match DART buoy timeseries as well as just the arrival times. The abstract is clear and accessible as is the start of the paper. The conclusions are fine too but some of the interesting supplementary work is only mentioned in the summary and then only briefly - it would be good for the implications of this to be more considered.

We completely take the point of the reviewer (in fact, in line with the previous reviewers) that our original submission needed significant improvements in terms of language and clarity and fluidity of the text. Again, we worked hard to improve these aspects and hope the revised submission adequately addresses these potential shortcomings.

Reference: previous literature cited appropriately.

Inflammatory material: none.

General comments:

1. Consistency of language. Authors refer to shock wave, acoustic-gravity waves, atmospheric waves, air-pressure jumps and similar terms - are these all the same thing? Are they subtly different? If so how? Ideally pick one term at the beginning and stick with it. You have some great material in this but could do a better job of bringing it all together in the end and explaining the implications to what was seen at HTHH.

R1: We take the point of the reviewer and made sure the text is now consistent in the terms employed.

Minor comments:

2. line 39 It is probably more to do with the area over which the displacement occurs rather than the total volume that determines the ability to become a trans-Pacific tsunami.

R2: Changed to “enough water” to avoid unclarity.

3. line 44 Unclear what you are saying is the necessary but not sufficient condition. Is this just the size of the explosive pressure? Is it the acoustic gravity waves? What would be sufficient? This does not make that clear.

R3: The text has been edited to clarify this statement. The paragraph now reads: “Large and violent explosive eruptions may trigger atmospheric perturbations by a sudden ejection, at supersonic speeds, of lava and gases into the air. If the explosive pressure is sufficiently large, the thrusting of volcanic products into the atmosphere can produce atmospheric acoustic-gravity wave modes travelling in air. [...]. Critically, acoustic-gravity wave fronts propagating over the ocean can force the sea surface to generate tsunami-like waves, known as volcano-meteorological tsunamis.”

4. line 49/84 and elsewhere Capitalise Plinian/Surtseyan etc.

R4: As suggested by reviewer #1, these volcanological terms were removed from the text, as they may represent unnecessary jargon. Therefore, we have substituted them by simple expressions that are comprehensive to the general audience (e.g. substituting “Surtseyan” by “shallow-water explosions”).

5. line 50 vulcanian? maybe volcanic.

R5: Corrected.

6. line 57 - refs to Pinatubo and Taupo - no refs for Krakatau or Kamchatka.

R6: References added.

7. line 80 query does the edifice just refer to the small cone that formed between Hunga Tonga and Hunga Ha'apai? The underlying sub aerial part has been there for far longer.

R7: The reviewer is right, and we referred to the volcanic cone as edifice. Now, the correct terminology (“volcanic cone”) is used as suggested by the reviewer. In order to make the text more pithy and objective, this section was also significantly re-written, and unnecessary details were removed to focus the text on what is essential.

8. line 84 Hunga volcano has produced...

R8: Corrected.

9. line 90 intercalated - is this a technical term? it certainly isn't a common term.

R9: Term changed to “alternated”.

10. line 96-98 centre of the new cone had already been destroyed before the Jan 15 eruption (probably the Jan 14 eruption as satellite image pre-eruption shows the middle missing).

R10: We agree with the reviewer and we removed this sentence.

11. line 104 remove simultaneously.

R11: Removed

12. line 105 metric-sized is meaningless - do you mean tsunami waves on the order of a metre high? (Amplitude or peak to trough?)

R12: Replaced by the numeric values of observed tsunami wave amplitudes at tide gauges.

13. line 121 common tsunamis - better to just say earthquake triggered tsunamis. Could also just say the shallow water wave limit as it has nothing to do with the wave generation but means that they had to be forced waves.

R13: As suggested by the reviewer, "common tsunamis" changed to "earthquake-triggered tsunamis".

14. line 126-129 - this seems cumbersome - can you not just indicate the shallow water wave speed.

R14: Rephrased accordingly, now it reads: "*Our analysis suggests that the Tonga tsunami propagated at a speed of ca. 1000 km/h, showing no significant decrease from deep to shallow water, as expected for earthquake-triggered and point-sourced tsunami waves as their phase speed (c) is linearly dependent of the water depth (h) over which they propagate (i.e., $c = \sqrt{gh}$, where g is the gravity acceleration).*" (See pages 10-11 of the annotated MS)

15. line 135 figure only shows 1.5-2 days.

R15: Corrected to "...more than 1.5 days"

16. line 136 'resulting wave amplitudes over the oceans' this needs to be rephrased. Also worth noting that sea level gauges only tell some of the story and run-up heights in some locations in Tonga were significantly higher.

R16: Rephrased accordingly.

The suggestion on the local tsunami impact is also implemented into the Section on the "Exceptional tsunami followed the Tonga-Hunga Ha'apai volcanic explosion" once we mentioned the tsunami wave record at Tonga Island "*At a distance of ca. 67 km, the tide gauge of Tonga Island recorded a maximum wave amplitude of 1.14 m (Fig. 2b, Supplementary Information Table 1) that, however, does not reflect the full near-field tsunami impact as high runup heights (ca. 15 m) were also observed in Tongatapu Island, Tonga Archipelago, suggesting the contribution from a local tsunami source (likely of volcanic or gravitational origin).*" (See page 11 of the annotated MS).

17. Fig 2a - Please colour code contours with the same scale as the dots - this will make it far easier to see the differences.

R17: Fig 2a was colour coded as suggested by the reviewer.

18. Fig 2b in caption refer to height but amplitude in figure - make sure it is clear what is being plotted.

R18: The caption was corrected accordingly as what is being plotted are the tsunami wave amplitudes.

19. Fig 2c Sea level records shown here do not match with heights given in Fig 2a - obviously taken from other locations. This doesn't invalidate what you are saying but needs to be explained. Also, are the coloured bands calculated from an external model or are these just highlighting where the observed disturbances occur?

R19: Indeed, the maximum wave amplitudes extracted from the sea records shown in Fig. 2c are plotted in Fig 2b. The coloured bands, on the other hand, are just highlighting where the observed disturbances occur, i.e. during each of the passages of the acoustic-gravity waves as they travelled across the globe; this is now explained in the Figure caption.

20. Fig 2d Underlying picture is brightness temperature which is not explained in the caption. The caption says that this shows good agreement with actual barometric pressure anomalies but how this is shown by this figure is not obvious (if this is where the bands in Fig 2c are calculated from then state that).

R20: Thanks for spotting that, the figure caption was updated with the explanation on the brightness temperature. Also, we explained that there is a good agreement in the times of the passage of the AGW (red contours in Fig. 2d) and the occurrence of the barometric pressure anomalies (red bands in Fig. 2c)

21. line 153 remove 's

R21: Removed

22. line 159 It shows they are correlated, I am not sure it shows they are intrinsically linked.

R21: changed to "correlated"

23. line 163 extended data fig 3 does not show extra atmospheric waves - these can be seen in the animations.

R23: "Supplementary Information Animation 1" called here

24. line 168 Is this really coupled? From what you have said it sounds like you reproduce the atmospheric disturbance by assuming it spreads radially around the earth and reduces in height. This is used to force the tsunami model. I don't consider that coupled.

R24: We agree with the reviewer's criticism and changed throughout the text "couple numerical model" to "forced tsunami numerical model"

25. line 173 'our numerical modelling of a tsunami forced by ...' Is this the same thing as the coupled model you mention above (it is a far more accurate description of it but the way you say it sounds like it is something different.

R25: Now homogenised throughout the text.

26. line 176-177 I understand what you are saying but I think it would be better to state that it can be rapidly generated or something like that which make it appear to "jump".

R26: The reviewer's suggestion has been implemented and we also re-worked Fig. 2b to support such a statement. The sentence now reads: *"Moreover, it provides a valid mechanism to explain the interoceanic wave propagation, showing waves rapidly generated across the oceans, which make them appear to "jump" from one ocean to another (e.g., from the Pacific to the Atlantic Ocean across Central America) with minimal loss of wave height (Fig. 3b; Supplementary Information Animation 2)"* (See page 16 of the annotated MS).

27. line 179 further excited

R27: Corrected.

28. 180-183 Point to details in methodology?

R28: Done

29. line 183 You miss the most important part of this - that your model actually reproduces the time series well at many locations

R29: We thank the reviewer for calling our attention to emphasize such an important result of this work. To do so, we added a sentence about this result.

30. 185 What do you mean by overall maximum uniform wave heights distribution

R30: Rephrased accordingly

31. line 191 Fig 3 incorrect terminology - its not the ocean swell that it is resonating with it is the tsunami waves.

R31: Corrected accordingly.

32. 195-196 I can't see evidence that the maximum wave heights are in agreement with observations - why don't you plot the DART buoys on 3c in the same colour scheme as the modelling - this would show whether that claim is true or not.

R32: Fig 3c (Fig 3d in the revised version of the MS) is improved to allow comparison of modelling with data. DART buoys' recorded maximum wave amplitudes were accordingly plotted in the same colour scheme as the modelling.

33. line 198 What is the novel model? meteotsunamis are already known about. Far field wave from Krakatau have been shown to be from this mechanism - you even quote studies that look at Taupo etc in this light.

R33: We take the point of the reviewer and, accordingly, removed the word "novel" from the description of the proposed model.

34. line 204-206 What is your atmospheric model based on? Are you actually modelling this or just giving an explanation? I really don't follow what you mean by 'distribution of sources over the volcanic column' it sounds like you are talking vertically but then you are saying this is the air-sea interface. Isn't it more to do with refraction and wave speed at different heights? I am not sure how diffraction comes into it.

R34: Yes, both refraction and variation of densities in air result in different tangential speed at different heights. In addition, here we wanted to emphasise that as the flow leaves the circular cross-section of the water layer to penetrate to the air layer, this can be pictured as flow through a circular opening which results in the diffraction of waves. In potential flow this would be described by distribution of sources over the cross-section of the cylinder, thus there is a component of tangential velocity as well, as the flow leaves the water and enters the air layer. We have modified and clarified the wording as follows: "*At the air-water interface, the explosion can be pictured as a flow through a circular opening or a distribution of sources over the volcanic column cross-section. Thus, due to wave diffraction (Huygens-Fresnel principle), and density stratification of the air and refraction, the generated acoustic-gravity waves propagate mainly tangentially to the ocean surface.*" (See page 21 of the annotated MS).

35. line 207 At these scales 'unperturbed' is irrelevant.

R35: 'unperturbed' was deleted

36. lines 207-209 The point is that it is a forced wave. High pressure oscillation doesn't really mean anything

R36: Indeed, text has been modified accordingly.

37. line 212 second tsunami wave train - make sure you are clear what you mean by this -

earlier you were attributing this to multiple passes of the AGW, again sloppy use of multiple terms. When you say existing tsunamis do you mean the tsunami wave train first generated by the AGW or do you mean the original local tsunami.

R37: For the sake of clarity, we updated Fig. 3b to support our explanation of 1st and 2nd wave trains as well as “new” tsunami generation. We also harmonised and edited the text accordingly.

38. line 217-218 Incorrect terminology, it is not the long swell - swell refers to deep water waves - these are shallow or intermediate water waves. Also this whole part suggests that this is a different mechanism to the previous paragraph (212-216) I don't see how this is different.

R38: Thanks for spotting that, the terminology is corrected throughout the text and “swell” is replaced by “long-waves” or simply “tsunami waves”.

We also agree that the part on resonance mechanism is similar what is said in the previous paragraph, and we edited the text to avoid such a repetition.

39. line 219-221 It is not just significantly inhibited - for resonance to occur the speeds need to be at least in the same ballpark and ideally the same speed... You are just repeating yourself here.

R39: We agree with the reviewer's comment and we changed the text accordingly: “*This resonance mechanism is, however, unexpected at shallow waters, given that tsunami waves become slower and do not resonate with acoustic-gravity waves in these areas*” (See page 21 of the annotated MS).

40. line 222-223 Worth explaining why you do get tsunami waves in Caribbean/Mediterranean - is it only in the deeper places of these ocean basins?

R40: Explanation was added as follows: “In ocean basins, comprising both shallow and deep waters like in the Caribbean and Mediterranean, tsunami amplification is believed to occur in deep-water areas, as shown by the numerical model (Fig. 3c), then the amplified waves propagated towards the shallow water coastal areas reaching ca. 20-50 cm at some locations (Fig. 2b, Supplementary Information Table 1).” (See page 22 of the annotated MS).

41. lines 229-230 Pacific Islands (oceanic is a tautology here).

R41: We did not implement this correction, as the meaning here relates to all oceanic islands and not just Pacific Islands. By definition, oceanic islands are those that rise to the surface from the floors of the ocean basins, and this is the meaning intended here.

42. line 231-232 or seismic tsunamis which are linear sources and are the most common sort of tsunami (point source tsunamis are not that common).

R42: Corrected to “earthquake-triggered tsunamis”.

43. 233-234 It is more that they have energy being pumped in to increase their size. Also the atmospheric forcing has managed to excite relatively linear shallow water waves which do travel with minimal energy loss...

R43: We agree with the reviewer’s comment and we believe that her suggestion is covered by the statement: *“experience minimal energy dissipation as a function of increasing distance”*

44. line 233 also that enough energy gets pumped in that they can reach the size of seismic tsunamis.

R44: We agree and accordingly inserted the reviewer’s suggestion: *“can reach the size of earthquake-triggered tsunami as enough energy gets pumped into them”*

45. Fig 4 This fig suggests that 1, 2 and 3 are successive moments in time but in 2 there is already a tsunami some of which is reaching beyond the AGW. The first wave will be a bound wave pushed by the AGW and there will be a wave train behind it - later waves may be reinforced by later AGW.

R45: We updated the Figure 4 caption to avoid unclarity by adding explanation of the tsunami “already” present in 2. The following sentence was added *“In 2, the dashed blue signal corresponds to the resonant tsunami propagating under gravity towards the coast.”*

46. line 242 not ocean swells.

R46: The reviewer is right, and the terminology is corrected accordingly. Throughout the text “ocean swell” were replaced by “tsunami waves” or “ocean long-waves.”

47. line 246 analytical analysis (is there any other kind?).

R47: Corrected to “analytical model”

48. line 256-257 What evidence do you have that they are more common than previously thought? It seems like you might get a big one every hundred years or so (last big one being Krakatau).

R48: The reviewer is right, and we removed this sentence.

49. lines 339-340 'constraint its origin' 'oceanic data concern sea-level accurate records' - these do not make sense.

R49: Rewritten.

50. 342 complemented? It would take a lot more to complete.

R50: Replaced by “enriched”

51. 344 concerns - do you mean consists of?.

R51: Replaced by “consist of”

52. 346 specifics.

R52: Corrected

53. 352-353 It is not common to talk about data not being harmed - affected would be a better word. What do you mean by data recalculated to millimetres. In fact right through to 356 does not really make sense and needs to be rewritten.

R53: All the section was rewritten.

54. 366 peak to peak wave height? I assume you mean peak to trough (or vice versa) you are not going to get the maximum wave height by measuring peak to peak?.

R54: Corrected to “crest-to-trough”

55. 367-369 This seems like assigning numbers for the sake of seeming rigorous without adding anything - You could just say 'we only kept arrival times where there were only minimal doubts about the arrival (or something like that).

R55: Rephrased as suggested by the reviewer.

56. 374 Also this whole section shows that you don't have a coupled model - you empirically estimate the pressure forcing (which is completely off in the nearfield as can be seen by looking at the BoM data from Nuku'alofa but does a good job in the far field)

R56: As stated in our previous answer to the referee's criticism on the coupled model, we agreed and changed accordingly throughout the text.

57. line 431 - limited rather than constrained

R57: Corrected.

Analytical solution

58. This is some interesting analysis but this entire section is very sloppy and not consistent in the terms used - so while the overall analysis could be valid it is poorly presented and so difficult to ensure it is correct. This is a shame because it could be very interesting. Especially if it were better connected with the implications for HTHH. How well does this analytical model fit what was seen/modeled in the open ocean?

R58: We have revised this section entirely. To avoid inconsistency in the terms, each of the two presented models now has its own boundary conditions. We improved presentation of the figure and added further explanation – the figure now includes a top view to better describe interaction. We also added another figure that quantifies the acoustic pressure at Kaitaia, New Zealand to compare with dart buoy data. The figures are now discussed in more detail.

Some specific issues are given below:

59. Zeta appears to be the free surface in (7) but you've just said it is h . You haven't defined in this what you consider the free surface (the top of the atmosphere or the ocean surface)

R59: That's a typo and has been corrected. Each of the two models are now presented and defined independently, each with its own boundary conditions to avoid confusion.

60. 453-455 First you assume that the interface is rigid and then in (6) $z=-1/\mu$ which is below where you are considering (assuming z positive up - not necessarily a given here...)

R60: Thanks for spotting that. That is a mistake (related to the previous responses) when trying to reduce the number of equations used. We have corrected the mistake by adding boundary conditions for each of the two models separately.

61. 455-457 Tau is half the duration in your example

R61: Indeed, thanks for spotting that, the typo was corrected.

62. 459 the equations state $z=zeta$ but you say they are both on $z=0$ - these can't both be true. In the text you talk about $z=h$... zeta is never mentioned in the text but I assume this is the free surface - this seems to just be copied from ref 32 as is using the same terms. Equation 10 cannot be obtained by expanding equation 7.

R62: All corrected now, and we have added the proper boundary conditions for each of the models independently, and explained that these are after linearisation so there is no need to mention the surface elevation ζ (or $\zeta+h$) anymore.

63. 489 typo

R63: typo was corrected, and proper maths presentation is applied.

64. 496 - inconsistent with reference paper

R64: typos corrected – thanks.

65. ED Fig5 A better explanation of what is being shown would be useful. The top seems to be the Gaussian pressure disturbance (and I can just make out A on the vertical axis) but is that the SGW solution also plotted on it? I think it is useful both being plotted on the same graph and then the SGW being plotted again magnified below but this needs to be stated clearly. Also these are solutions for radius r but you are plotting positive and negative r... Is there any physical significance to the negative solutions? If not these should not be plotted. You identify earlier tsunami amplitude amplified due to AGW but this appears to be only in the negative r domain so may be unphysical.

R65: thanks for the comment. In the original figure “r” represents the propagation axis, which is now replaced by X to avoid confusion. The point X=0 is some distance away from the source, where the acoustic-gravity wave first impacts the interface (the calculation is initiated). So in that regard, (+X) means propagating away from the source, whereas (-X) means getting closer to the source – still in the outer field. So both signs are physical. Note that the top figure shows the amplitude evolution of the AGW only, and the lower figure shows the SGW only. We have updated the figure quality and added more detailed explanations.

Reviewer Reports on the First Revision:

Referees' comments:

Referee #1 (Remarks to the Author):

I'm amazed at how thoroughly and quickly the authors responded to not only my comments but those of the other two reviewers. I think that the new manuscript is about 1000% improved and is now a clear, cogent paper that will be very well received by a wide readership. I hope that it is published soon. Congratulations in advance.

All the best,
Matthew Alford

Referee #2 (Remarks to the Author):

Dear authors,

My opinion on the initial version of the manuscript was already positive, but this new version is much improved. It seems to me that all the comments of the reviewers have been taken into account. Some parts of the text have been completely rewritten, thus making the text more attractive and more accessible for all readers. Some points have been clarified (e.g. definition of acoustic-gravity waves, legends and figures, distinction between the effects of the tsunami in the near field or in the far field, etc.).

Figure 1 of the extended data is very useful and convincing. It could even come with the main text (to be discussed with the Editor).

In conclusion, I find that the manuscript is now ready for publication and I again congratulate the authors for this work which will be very useful to our community and also very interesting for a wide audience.

Best regards
Raphaël Paris

Referee #3 (Remarks to the Author):

The authors have done a good job of responding to most of my comments about their manuscript and the revised version is substantially improved. I now only have a few, relatively minor, comments that I would like to see addressed in this manuscript.

My previous comments on Key results and originality still stand:

Key results: Explanation of the far reach and long duration of the far-field tsunami generated by the Hunga Tonga - Hunga Ha'apai eruption - really good agreement in timing and time series with open ocean DART buoys, analytical solution of the resonant interaction between the acoustic-gravity wave

(AGW) and the surface gravity waves (SGW)

Originality: As the authors state, this is the first time that we have had data of this sort from an eruption of this magnitude and so understanding how the far field reacts is important and original. The analytical work synthesises earlier work (appropriately referenced) and extracts the parts that are pertinent in this case. This will be of great and immediate interest to people in my field as well as other adjacent fields.

Aside from the relatively minor comments below I am happy with the validity, data and methodology, clarity and context, conclusions, references etc.

More major comments:

Consistency of terms:

The following terms are all used in the manuscript at different points.

air-pressure disturbance, pressure jump, air pressure wave front, air wave, acoustic gravity wave, high frequency atmospheric disturbances,...

I could be convinced that there are two separate concepts - the acoustic gravity wave and the air-pressure disturbance that marks the passing of the wave. I suggest that using any more terms than this is just muddying the waters. I would pick two consistent terms for these two concepts and stick with them. If there are more subtle differences between these other terms and you are using them because there is a difference you need to make it clear what those differences are.

Nearfield pressure:

In 439-440 you state that there is no barometric pressure readings at source. While there are none 'at source' the BoM reading from Nuku'alofa is readily available. The point is that the near-field pressure is actually quite different from the far field pressure which is why you don't use it (I assume). This needs to be acknowledged. Because the focus of the paper is on the far-field tsunami I don't see this as a significant problem (as long as it is acknowledged) and it makes sense to use the Kaitaia, NZ record as being relatively close but the far-field signal.

Minor comments:

line 18 remove comma after tsunamis

line 63 far-reaching volcanic tsunamis [I would recommend including volcanic in this as seismic tsunamis can have global reach so that is not as exceptional]

line 84 - I would recommend rewording of "constrain the actual range of physical model parameters" What we have is an example of this phenomenon occurring but it far more information would be required to limit the range of parameters that are valid for this phenomenon.

line 144 no significant decrease in speed for deep to ...

line 146/147 c is dependent on depth but it is not linearly dependent on depth as you show in the next line $c = \sqrt{g \cdot h}$

line 174/175 make it clear that the dots are the observed travel times at stations and the background colour is the point source tsunami travel time

line 184/395 you say deg C in caption and Kelvin in text. I realise the difference is minor but be consistent.

line 444/447 wrong reference - it should be 33

line 447-450 you don't define p_0 here. Also the explosion energy is not obviously used anywhere in the analysis that I can see - if it is then make the connection more explicit.

Eqn (1) and later equations - R used for earth's radius in first and R_0 in later eqns

Eqn (2), (3), (4) $\sin(d/R_0)$ - I suspect that this is the same as $\sin(\phi)$ in eqn (1) - worth at least acknowledging that these later sine functions are assuming radians given that you specify ϕ is in degrees (could just change ϕ to d/R_0 for simplest consistency)

line 464 these need to be subscripts to be consistent with equations: R_0 , C_1

Eqn (9) where did the ρ come from? there is no ρ in earlier equations

line 681 3.4×10^{23} ergs - Also I see this where you are using the estimate of the volcanic energy. It is worth mentioning somewhere around line 450 that this estimate is used in the point source tsunami model. It could also be mentioned around line 169 where you mention modelling a comparable point source tsunami.

ED Fig 3/4 minor point but it would be nice to have the full location names not just the short versions.

There are still a few times where the

e.g.

line 20/21 can cause ...

line 45 is possible through..., as illustrated by ...

line 67/68 still remain in our detailed knowledge...

line 252 i.e. the first tsunami wave explains the earlier ...

line 267 ... is , however, not expected in ...

line 271 Crucially, the highest tsunami ...

line 272 ..., as occurs at ...

line 275 This explains why ...

lines 279/280 - be consistent in use of tense... either "... to have occurred in ... waves propagated ..." or "... to occur in ... waves propagate ..."

line 385 exceptionally doesn't quite seem like the right word here

line 386 affected the atmosphere

line 433 "minimal doubts" I am not quite sure what you mean here - do you mean very little uncertainty?

line 459 Let d_0 be the ...

line 499 particularly in terms of ...

Author Rebuttals to First Revision:

RESPONSE TO COMMENTS FROM THE REVIEWERS

Referee #1:

I'm amazed at how thoroughly and quickly the authors responded to not only my comments but those of the other two reviewers. I think that the new manuscript is about 1000% improved and is now a clear, cogent paper that will be very well received by a wide readership. I hope that it is published soon. Congratulations in advance.

All the best,

Matthew Alford

R: We are delighted to hear that the reviewer is satisfied with the modifications to the manuscript, and we thank the reviewer for his extremely helpful comments that helped to improve the manuscript.

Referee #2:

Dear authors,

My opinion on the initial version of the manuscript was already positive, but this new version is much improved. It seems to me that all the comments of the reviewers have been taken into account. Some parts of the text have been completely rewritten, thus making the text more attractive and more accessible for all readers. Some points have been clarified (e.g. definition of acoustic-gravity waves, legends and figures, distinction between the effects of the tsunami in the near field or in the far field, etc.).

Figure 1 of the extended data is very useful and convincing. It could even come with the main text (to be discussed with the Editor).

In conclusion, I find that the manuscript is now ready for publication and I again congratulate the authors for this work which will be very useful to our community and also very interesting for a wide audience.

Best regards

Raphaël Paris

R: We are delighted to hear that the reviewer is also satisfied with the modifications to the manuscript, and we also thank the reviewer for his extremely helpful comments that helped to improve the manuscript.

Referee #3:

The authors have done a good job of responding to most of my comments about their manuscript and the revised version is substantially improved. I now only have a few, relatively minor, comments that I would like to see addressed in this manuscript.

My previous comments on Key results and originality still stand:

Key results: Explanation of the far reach and long duration of the far-field tsunami generated by the Hunga Tonga - Hunga Ha'apai eruption - really good agreement in timing and time series with open ocean DART buoys, analytical solution of the resonant interaction between the acoustic-gravity wave (AGW) and the surface gravity waves (SGW).

Originality: As the authors state, this is the first time that we have had data of this sort from an eruption of this magnitude and so understanding how the far field reacts is important and original. The analytical work synthesises earlier work (appropriately referenced) and extracts the parts that are pertinent in this case. This will be of great and immediate interest to people in my field as well as other adjacent fields.

Aside from the relatively minor comments below I am happy with the validity, data and methodology, clarity and content, conclusions, references etc.

R: We are delighted to hear that the reviewer is also satisfied with the modifications to the manuscript, and we also thank the reviewer for her extremely helpful comments – including these last edits – that helped to improve the manuscript.

More major comments:

1. Consistency of terms:

The following terms are all used in the manuscript at different points. air-pressure disturbance, pressure jump, air pressure wave front, air wave, acoustic gravity wave, high frequency atmospheric disturbances,...

I could be convinced that there are two separate concepts - the acoustic gravity wave and the air-pressure disturbance that marks the passing of the wave. I suggest that using any more terms than this is just muddying the waters. I would pick two consistent terms for these two concepts and stick with them. If there are more subtle differences between these other terms and you are using them because there is a difference you need to make it clear what those differences are.

R1: We take the point of the reviewer and in the revised version of the manuscript we use the two suggested concepts – the acoustic gravity wave and the air-pressure disturbance – throughout the text. The other terms (i.e., pressure jump, high frequency atmospheric disturbances ...) were also changed accordingly to make sure the text is now consistent in the terms employed.

2. Nearfield pressure:

In 439-440 you state that there is no barometric pressure readings at source. While there are none 'at source' the BoM reading from Nuku'alofa is readily available. The point is that the near-field pressure is actually quite different from the far field pressure which is why you don't use it (I

assume). This needs to be acknowledged. Because the focus of the paper is on the far-field tsunami I don't see this as a significant problem (as long as it is acknowledged) and it makes sense to use the Kaitaia, NZ record as being relatively close but the far-field signal.

R2: We agree with the reviewer and accordingly explained the reason behind using the New Zealand air-pressure data. The edited sentence now reads: *“As our work focussed on the far-field, we reconstructed the air-pressure disturbance around the Hunga Tonga-Hunga Ha'apai (HT-HH) volcano from barometric signals recorded at some distance from the source, i.e. the high-resolution (1 min) pressure data available in New Zealand.”*

Minor comments:

line 18 remove comma after tsunamis

Done

line 63 far-reaching volcanic tsunamis [I would recommend including volcanic in this as seismic tsunamis can have global reach so that is not as exceptional]

Done

line 84 - I would recommend rewording of "constrain the actual range of physical model parameters" What we have is an example of this phenomenon occurring but it far more information would be required to limit the range of parameters that are valid for this phenomenon.

Reworded accordingly.

line 144 no significant decrease in speed for deep to ...

Done

line 146/147 c is dependent on depth but it is not linearly dependent on depth as you show in the next line $c = \sqrt{g \cdot h}$

“linearly” removed from the sentence.

line 174/175 make it clear that the dots are the observed travel times at stations and the background colour is the point source tsunami travel time

Done

line 184/395 you say deg C in caption and Kelvin in text. I realise the difference is minor but be consistent.

The text in line 395 (i.e., Kelvin) refers to Fig. 1 (where the top-of-the-atmosphere brightness temperature is shown in Kelvin) not to Fig.2d where we plotted the difference in the brightness temperature. To clarify: we analyzed absolute top-of-the-atmosphere in °K (as expressed in Fig 1), but for the difference we used °C. Since this is a difference, and given that the change in one unit is the same in both scales, the difference is the same whether we use °C and ° K. We corrected accordingly the legend and caption of Fig2d.

line 444/447 wrong reference - it should be 33

Corrected to 33.

line 447-450 you don't define p_0 here. Also the explosion energy is not obviously used anywhere in the analysis that I can see - if it is then make the connection more explicit.

Suggestion implemented.

Eqn (1) and later equations - R used for earth's radius in first and R_0 in later eqns
Changed to "R_0" in equation 1.

Eqn (2), (3), (4) $\sin(d/R_0)$ - I suspect that this is the same as $\sin(\phi)$ in eqn (1) - worth at least acknowledging that these later sine functions are assuming radians given that you specify ϕ is in degrees (could just change ϕ to d/R_0 for simplest consistency)
The reviewer is indeed correct – changed accordingly

line 464 these need to be subscripts to be consistent with equations: R0, C1
Done

Eqn (9) where did the rho come from? there is no rho in earlier equations
Indeed, that was incorrect, and rho shouldn't appear in this equation. We corrected the eq9 by removing rho.

line 681 3.4×10^{23} ergs - Also I see this where you are using the estimate of the volcanic energy. It is worth mentioning somewhere around line 450 that this estimate is used in the point source tsunami model. It could also be mentioned around line 169 where you mention modelling a comparable point source tsunami.
We take the reviewer point and mentioned the use of the energy estimate as: *"This energy estimate is then used to simulate the tsunami generation from a point-source model (Extended Data Fig. 2a)."*

ED Fig 3/4 minor point but it would be nice to have the full location names not just the short versions.

Implemented as suggested. All full location names were added to the figure, as well as keeping the short names by which those stations are known in databases.

There are still a few times where the

e.g.

line 20/21 can cause ...
Done

line 45 is possible through..., as illustrated by ...
Done

line 67/68 still remain in our detailed knowledge...
Changed simply to "remain"

line 252 i.e. the first tsunami wave explains the earlier ...
Done

line 267 ... is , however, not expected in ...

Done

line 271 Crucially, the highest tsunami ...

Done

line 272 ..., as occurs at ...

Done

line 275 This explains why ...

Done

lines 279/280 - be consistent in use of tense... either "... to have occurred in ... waves propagated ..." or " ... to occur in ... waves propagate ..."

Done

line 385 exceptionally doesn't quite seem like the right word here

Word "exceptionally" removed from the text, as it is not necessary

line 386 affected the atmosphere

Done

line 433 "minimal doubts" I am not quite sure what you mean here - do you mean very little uncertainty?

Corrected to "very low uncertainty"

line 459 Let d_0 be the ...

Corrected

line 499 particularly in terms of ...

Corrected